# Resisting Memorization-Based APT Attacks Under Incomplete Information in DDHR Architecture: An Entropy-Heterogeneity-Aware RL-Based Scheduling Approach

**DOI:** 10.3390/e27121238

**Published:** 2025-12-07

**Authors:** Xinghua Wu, Mingzhe Wang, Xiaolin Chang, Chao Li, Yixiang Wang, Bo Liang, Shengjiang Deng

**Affiliations:** 1Beijing Key Laboratory of Security and Privacy in Intelligent Transportation, Beijing Jiaotong University, Beijing 100044, China; 22115141@bjtu.edu.cn; 2Institute of Computing Technology, China Academy of Railway Sciences Corporation Limited, Beijing 100081, Chinaliangbo713@126.com (B.L.);; 3The First Research Institution of Ministry of Public Security, Beijing 100006,China

**Keywords:** DHR architecture, entropy, FlipIt game, mimic defense, reinforcement learning

## Abstract

The rapid advancement of artificial technology is giving rise to new forms of cyber threats like memorization-based APT attacks, which not only pose significant risks to critical infrastructure but also present serious challenges to conventional security architectures. As a crucial service information system in railway passenger stations, the Railway Passenger Service System (RPSS) is particularly vulnerable due to its widespread terminal distribution and large attack surface. This paper focuses on two key challenges within the RPSS Cloud Center’s Double-Layer Dynamic Heterogeneous Redundancy (DDHR) architecture under such attacks: (i) the inability to accurately estimate redundant executor scheduling time, and (ii) the absence of an intelligent defense scheduling method capable of countering memorization-based attacks within a unified and quantifiable environment. To address these issues, we first establish the problem formulation of optimizing defender’s payoff under incomplete information, which applies information entropy of DDHR redundant executors to reflect attacking and defending behaviors. Then a method of estimating attacking time is proposed in order to overcome the difficulty in determining scheduling time due to incomplete information. Finally, we introduce the PPO_HE approach—a Proximal Policy Optimization (PPO) algorithm enhanced with quantifiable information Entropy and Heterogeneity of DDHR redundant executors. Extensive experiments were conducted for evaluation in terms of the two entropy-related metrics: information entropy decay amount and information entropy decay rate. Results demonstrate that the PPO_EH approach achieves the highest efficiency per scheduling operation in countering attacks and provides the longest resistance time against memorization-based attacks under identical initial information entropy conditions.

## 1. Introduction

The Railway Passenger Service System (RPSS) is a critical information system that integrates and manages service equipment such as station announcements and passenger guidance. In recent years, with the continuous maturation of cloud-native technology applications in railway information systems, the railway passenger service system has also gradually shifted toward centralized deployment in regional cloud centers. While the centralized deployment model in cloud centers facilitates rapid integration of station systems across different cities, it also increases the risk of various types of attacks. In particular, APT attacks targeting critical infrastructure such as transportation, power, automotive, and technology sectors [1,2] have been on the rise, leading to an escalation of operational risks in station operations.

With the rapid development of Artificial Intelligence (AI) technology, the number of APT attack methods based on AI technology has also been increasing. The characteristics of APT attacks—targeting, stealth, and persistence—have been further enhanced. For example, they exhibit memory and deep mimicry capabilities, meaning attackers can use covert detection methods to gradually gather information about the defender’s defensive resources and strategies, accumulate memorization-based experience, and then deeply mimic normal traffic and business behavior, continuously adjusting their attack strategies in real-time [3] to achieve more efficient attacks.

Currently, the use of the kill chain model [4], ATT&CK knowledge base technology [5], and intrusion detection technologies based on neural network models such as CNN, RNN, and GNN [6,7,8] has achieved some success in detecting APT attacks. However, for RPPS systems deployed on railway intranets, such passive defense methods still have the following two issues:(i)Existing methods remain inadequate for detecting and identifying APT attacks. While current detection technologies have made significant strides in accuracy, speed, and reducing sample dependency, they still largely rely on prior knowledge. This makes accurate detection of sophisticated attacks—such as memory-resident APT attacks that exploit zero-day vulnerabilities—particularly challenging. Furthermore, attack strategies can adapt in real-time to the defense landscape, rendering post-detection countermeasures highly difficult. Another critical issue is the lack of redundancy in defense strategies. A missed detection can cause system compromise and then chaos at passenger stations.(ii)Traditional defense strategies centered on attack identification incur high costs throughout their deployment lifecycle, rendering large-scale investment in a single system economically challenging. The inherent stealth of APT attacks means that available attack samples targeting railway internal networks are scarce, while the process of creating such samples is itself resource intensive. Moreover, the operational constraint characteristics of industrial control network deployments further elevate the maintenance costs associated with real-time updates to attack sample databases. As a result, substantial investment in a tailored defense system for an individual platform is impractical.

To mitigate the challenges outlined above, a strategic solution lies in deploying a defense architecture with a constellation of desirable characteristics: high redundancy, inherent fault tolerance, low capital expenditure, and deployment flexibility. This core architecture may be further enhanced by integrating real-time dynamic detection mechanisms, as project conditions dictate. Presently, predominant architectures that fulfill these requirements and are amenable to centralized cloud deployment encompass DHR (Dynamic Heterogeneous Redundancy) architecture [9,10], fault-tolerant architecture [11,12,13], and redundant-class architectures [14,15,16,17], among others. The characteristics and differences in these technologies are summarized in Table 1.

Clearly, compared to other defense architectures, the DHR architecture is well suited to the security construction of RPSS due to its high redundancy capabilities, fault tolerance that does not rely on any single component’s failure, and compatibility with centralized deployment.

The existing studies of mimic defense technology have demonstrated its capability to resolve fundamental security challenges. However, when dealing with memorization-based APT attacks (referred to as memorization-based attacks) deployed in railway passenger station cloud environments, the following two weakness remain.

Weakness 1: In real-world cloud environments, memorization-based attacks are often difficult to detect in real time, leading to delayed execution of redundancy scheduling strategies. Existing studies lack designs for such scheduling mechanisms under these specific attack scenarios.

Weakness 2: Memorization-based attack strategies often adapt to changes in defense strategies, and existing research lacks adaptive defense methods that can dynamically adjust to such attack strategy changes.

This paper addresses the above issues by combining our previous research on the double-layer DDHR technology for multi-layer cloud networks in railway passenger stations [18]. The main innovations of this paper include the following:(1)We establish the FlipIt-game-based problem formulation of optimizing DDHR defender’s payoff under incomplete information. It applies information entropy of DDHR redundant executors to reflect attacking and defending behaviors, detailed in Section 3. By transforming the defense against memorization-based attacks into a problem of time allocation for public resources and changes in information entropy metrics, it achieves payoff quantification for redundant executor scheduling strategies against memorization-based attacks. In the DDHR architecture model, all redundant executors performing computations can be considered as public resources.(2)We propose a method of estimating attacking time in order to overcome the difficulty in determining scheduling time due to incomplete information, detailed in Section 3.2. This method first defines some special executor points (denoted as anchor in the rest of the paper) in order to introduce heterogeneity to DDHR redundant executors. By employing a predefined scheduling strategy for redundant executors with special heterogeneity and then estimating the attack time of the anchor point redundant executor, the difficulty of estimating the executor scheduling time is overcome. That is, this addresses the challenge of setting redundant executor scheduling times when memorization-based attacks remain undetected or detection is intermittent.(3)We propose the PPO_EH approach—a Proximal Policy Optimization (PPO) algorithm enhanced with quantifiable information Entropy and Heterogeneity of DDHR redundant executors. We not only detail PPO_EH including Markov decision process of the scheduling problem and the scheduling steps in Section 4.2 and Section 4.3 but also present the DDHR scheduling framework deploying PPO_EH in Section 4.1.

We summarize the research methodology of this paper as follows. First, in Section 3 after Section 2 of Related work, we formulate the optimization problem of the defender’s equilibrium payoff. This formulation is established by defining the FlipIt-game payoff model under imperfect information, along with the anchor-assisting heterogeneous redundant executor scheduling method and attack time estimation method. Then, in order to solve this problem, we develop the RL-based DDHR architecture redundant executor scheduling approach—PPO_EH—in Section 4. Section 5 conducts experimental simulations. Section 6 introduces the conclusions of this paper and future work prospects. Table 2 details the symbols to be used in the left paper.

## 2. Related Work

### 2.1. Research on APT Attack Defense Strategies for Integrated AI Technology

A typical APT attack can be considered a multi-stage attack, which can be roughly divided into the following five main stages [19], illustrated in Figure 1.

(1)Reconnaissance stage. Attackers use social engineering techniques and open-source intelligence tools to gather as much information as possible about the organization’s technical environment (such as routers, firewalls, etc.) and the background of key personnel (such as social activities, frequently visited websites, etc.) to identify potential attack vectors. Once intelligence gathering reaches the desired level, attackers begin planning attack strategies while preparing the necessary tools.(2)Establishing a foothold. Attackers bypass system defense using various technical means such as malware, web application vulnerabilities, and spear phishing, while establishing command-and-control (C&C) channels to deploy subsequent attack operations.(3)Lateral movement. Attackers hide within legitimate traffic and normal behaviors to avoid detection while further expanding their control privileges.(4)Data exfiltration. Attackers seize opportunities to steal confidential information or disrupt target systems. Once the target attack is successful, attackers export data to remote servers via C&C channels.(5)Disappearance. When attackers obtain the desired data, they may choose to terminate the attack or continue to lie dormant. Furthermore, recent studies indicate that attackers may attempt to erase attack traces after a successful attack, such as deleting log records and clearing temporary files, to prevent their actions from being tracked and traced.

In recent years, with the rapid development of AI technology, attackers have been able to efficiently carry out any of the above attack steps using AI technologies such as WormGPT4.0, FraudGPT1.0, and deepfakes1.0. In 2023, AI-based deepfake fraud increased by 3000%, and AI-based phishing emails grew by 1000%. Currently, multiple APT organizations have been identified as using AI to carry out numerous cyberattacks, including notable groups such as Sidewinder, Mysterious Elephant, and Donot [20].

To counter AI-APT attacks, research on AI-based defense technologies targeting APT attacks is also advancing rapidly. In terms of attack detection technology, Patel et al. [21] and Tadesse et al. [22] proposed a technique that converts DDoS/DoS attack traffic features into images and then uses CNN for attack classification. Xu et al. [23] and Yin et al. [24] addressed the issue of detection training with limited data samples, respectively, proposing a detection method based on a CNN meta-learning framework and a multi-scale CNN combined with a bi-LSTM arbitration dense network model to achieve attack classification and identification. Sun et al. [25] proposed an innovative intrusion detection system (IDS) that combines data preprocessing with four deep neural network models: convolutional neural networks (CNN), bidirectional long short-term memory networks (BiLSTM), bidirectional gated recurrent units (BiGRU), and attention mechanisms to achieve accurate identification of network attacks. When evaluated using the NSL-KDD dataset, its effectiveness in multi-classification identification was validated. Yang et al. [26] proposed an innovative intrusion detection method called Hypergraph Recurrent Neural Network (HRNN), which enhances information representation capabilities by converting traffic data into hypergraph structures, while using recurrent modules to extract temporal features, achieving a combination of spatio-temporal features and improving the accuracy of APT attack detection. Hasan M et al. [27] proposed a method combining XGBoost machine learning with explainable analysis (XAI), which not only improves the prediction accuracy of APT attacks but also generates explainable and actionable prediction results using SHAP technology. Li et al. [28] addressed the false positive issue caused by benign data fluctuations in Provenance-based Intrusion Detection Systems (PIDSes) by proposing the MIRDETECTOR anomaly detection system, which evaluates features based on three dimensions: structural features, attribute features, and malicious intent. This system effectively reduces false positive rates and achieves efficient real-time detection.

In terms of attack defense, Chen et al. [29] proposed a framework for collaborative defense of multiple cloud networks based on a game model and introduced a group multi-agent reinforcement learning defense algorithm (GMADRLD) to achieve dynamic collaborative scheduling of defense strategies for multiple cloud networks. Huda et al. [30] addressed the issue of low operational efficiency when networked physical systems (CPS) face APT attacks, proposing a semi-supervised learning-based attack feature recognition method. They also integrated CPS system management nodes to achieve dynamic protection against malicious software and malicious attack traffic. Zhou et al. [31] addressed the vulnerability of edge nodes in industrial internet systems to APT attacks by proposing a self-evolving system model based on optimal control and intelligent edge game theory. They utilized a DQN network to achieve dynamic detection and defense strategy against APT attacks, thereby enhancing the edge nodes’ ability to resist attacks. V. Phan et al. [32] propose DeepAir, an adaptive intrusion defense response solution based on deep reinforcement learning. By deploying it in the SDN decision-making module, it achieves dynamic intrusion response, maximizing defense performance while minimizing the negative impact on the forwarding of benign traffic in the SDN data plane and the cost of strategy deployment. Cao et al. [33] addressed the issue of slow learning rates for unknown attacks in internal IoT networks by proposing a deep reinforcement learning (DRL)-based mobile defense (MTD) routing scheduling mechanism. It deploys specific honeypot nodes in the IoT network and uses a ResNet architecture for network state sensing to quickly capture suspicious traffic, then performs real-time learning to rapidly enhance protection against unknown attacks. Yoon et al. [34] addressed the issue that APT attacks on vehicle networks may directly impact defensive AI agents, proposing a DRL-based resource allocation and deployment framework—DESOLATER. This framework employs multi-agent collaborative management strategies and memorization-based reinforcement learning anomaly detection mechanisms, thereby improving defense against APT attacks.

In summary, mainstream AI-based defense or detection technologies are continuously advancing in addressing AI-APT attacks, becoming the primary means of countering APT threats. However, most of these technologies rely on the long-term accumulation of training datasets. Under private deployment conditions, they often face challenges such as rapid iteration of APT attack methods coupled with slow maintenance and updates, leading to a decline in protective capabilities; attacks targeting specific systems are highly targeted, and there are few discovered attack samples, resulting in insufficient defensive effectiveness.

### 2.2. Research on DHR Architecture Deployment and Scheduling Strategies in Cloud Environments

The DHR architecture model consists of input agents, execution entities, voters, policy schedulers, heterogeneous resource pools, and other components, as shown in Figure 2 [10]. Its basic processing flow is as follows: (1) The scheduling module dynamically allocates redundant executors from the redundant executors pool to the processing module based on a dynamic selection algorithm; (2) messages sent by users are forwarded by the input agent to different redundant executors within the processing module; (3) after processing the request, the redundant entities send the results to the voter, which makes a consistent decision and outputs the result; (4) redundant executors identified during the decision-making process as having inconsistent decisions are fed back to the scheduling module via a negative feedback mechanism for re-scheduling of redundant executors.

As the core architecture of mimic defense technology, the DHR architecture has unique characteristics such as dynamism, high redundancy, heterogeneity, and a computational decision-making mechanism, enabling it to maintain normal working capabilities even under conditions of intrusion. At the same time, the DHR architecture’s scheduling module, heterogeneous resource pool, and other configurations make it more suitable than other redundant architectures for centralized deployment in private cloud environments.

In recent years, there has been increasing research on DHR architecture in the deployment of various cloud network applications and the protection against unknown attack strategies. In terms of application deployment in cloud environments, Sepczuk [35] proposed a lightweight deployment device that combines DHR architecture with WAF firewalls in cloud environments. After being deployed on the core nodes of the system, it can detect abnormal traffic information in real time and dynamically establish temporary routing rules, thereby enhancing the system’s security. Wu et al. [36] proposed an active defense development framework (ICS) tailored for cloud-native environments, leveraging techniques such as multi-version assembly, multi-instance deployment, and diversified compilation to enhance the system’s ability to defend against unknown attacks. Wang et al. [37] address the issue of unknown security risks in IoT networks by proposing a trusted architecture DRLIA based on DDQN (Double Deep Q Network). This architecture models scheduling and decision-making processes using two layers of Q networks, thereby addressing the ability to respond to attack situations in real time. Li et al. [38] oriented to the connected automated vehicles (CAVs) system and network security, constructed an intelligent DHR scheme suitable for its characteristics using the CTMC model, and verified the feasibility of the architecture through simulation tests.

In terms of defense strategies against unknown attacks, Chen et al. [39] proposed a dynamic architecture assessment method based on game theory, treating the unknown attacker as an object in a long-term game. They used a Markov chain model to calculate and evaluate the payoffs of both the attacker and defender and guided the deployment of defense strategies based on the equilibrium solution between attack and defense. Shi et al. [40] proposed a mimetic adjudication strategy based on evolutionary game theory. By analyzing the evolutionary sub-strategies of each scheduling strategy, they identified game strategy factors related to heterogeneity and dynamically controlled the system scheduling cycle. Hu et al. [41] analyzed the heterogeneity of redundant executors and the probability of being attacked based on information entropy theory, proposed a defense chain model based on information entropy and heterogeneity, and finally conducted numerical analysis of attack success rates using a continuous Markov model to validate the effectiveness of the DHR architecture. Kang et al. [42] addressed the security issue of information leakage in redundant executors caused by unknown attacks by proposing a DP-DHR architecture based on differential privacy technology. They classified results using an advanced decision strategy and the hyper-sphere clustering algorithm, ensuring that even if the attacker breaches the system, the output of each actuator may still be correct.

In summary, the DHR architecture has achieved significant research and practical results in cloud environment deployment and countering unknown APT attack strategies. However, when faced with memorization-based attacks, the following two issues remain:(1)Existing research lacks studies on security defense architectures under conditions of incomplete attack information. Currently, research on active security defense architectures such as the DHR architecture typically assumes real-time detection of attack information when developing response strategies. However, in practical engineering scenarios, even with AI-assisted detection of APT attacks, numerous attack behaviors remain untraceable in real time. This prevents security defense architectures from effectively implementing their response strategies.(2)Research on developing intelligent defense strategies under unified, quantifiable environmental conditions remains scarce. Currently, studies on active safety defense architectures—such as DHR frameworks assisted by machine learning algorithms—often lack a standardized, quantifiable environment, resulting in methodological shortcomings in both training processes and performance evaluation.

Therefore, this paper addresses memorization-based attacks by optimizing the DDHR architecture under imperfect information conditions to estimate attack timelines, thereby providing a basis for formulating defense strategies. Simultaneously, it models the system within a unified information entropy metric framework and employs reinforcement learning algorithms to optimize and evaluate the scheduling strategy of the DDHR architecture.

## 3. Problem Formulation of Optimizing Defender’s Payoff Under Incomplete Information

This section presents the modeling of attack time estimation and defense payoff under memorization-based attacks with imperfect information. Section 3.1 establishes an equilibrium payoff model for the DDHR architecture under imperfect information and single-stage attack conditions, based on the FlipIt game strategy. In Section 3.2, addressing the information probing problem in memorization-based attacks under imperfect information, we design an optimization method for the DDHR architecture based on a heterogeneous redundant executor anchor placement strategy, along with a redundant executor scheduling strategy and attack time estimation method, thereby transforming imperfect information into perfect information. Section 3.3 presents the attack–defense equilibrium payoff model for the DDHR architecture in the information entropy environment.

### 3.1. Attack–Defense Payoff Model with Under Incomplete Information

We first construct an attack–defense payoff model based on FlipIt game for the DDHR architecture. As illustrated in Figure 3, the attacker’s payoff can be considered as continuously obtained during the time interval $t_{Ai}$ when the public resources are successfully occupied, and the defender’s payoff can be considered as continuously obtained during the time interval $t_{di}$ when the public resources are successfully occupied by the defender.

A 4-tuple Z = {N,T,A,U} is defined to denote the attack–defense payoff model, detailed as follows.

(1)$N\;=\;\{N_{A},N_{D},N_{o}\}$ $N_{o}$ includes the double-layer redundant executor set $\left\{N_{up},N_{low}\right\}$.(2)$T=\{T_{A},T_{D}\}$ represents the set of time intervals when both the attacking and defending sides occupy public resources. $T_{A}=\{t_{A1}\ldots t_{Ai}\}$ represents the set of attacking time, and $t_{Ai}$ represents the time when the i-th attack is successful and the attacking side occupies public resources. Similarly, $T_{D}=\{t_{D1}\ldots t_{Dj}\}$ represents the set of times for the defending side, and $t_{Di}$ represents the time when the j-th defense is successful and the defending side occupies public resources.(3)$A=\{A_{A},A_{D}\}$ represents the game strategy space, with the attacker’s strategy being $A_{A}=\{A_{A1}\ldots A_{A2m}\}$ and the defender’s strategy being $A_{D}=\{A_{D1}\ldots A_{D2m}\}$.$A_{Ai}$ represents an attack on redundant executor i, and $A_{Di}$ represents the defender actively scheduling a defense against redundant executor i.(4)$U=\{U_{A},U_{D}\}$ represents the payoff of the attacker and the defender, respectively.

We make the following 5 assumptions about model Z:

**Assumption** **1:**
*The total redundant executors of $N_{o}$ is 2m, and $N_{up}=N_{low}=m$.
*


**Assumption** **2:**
*The loss of revenue is identical for any upper-layer redundant executor that is compromised. Similarly, the loss of revenue is identical for any lower-layer redundant executor that is compromised.*


**Assumption** **3:**
*$N_{A}$ and $N_{D}$ obtain equal returns from occupying public resources for time t.*


Equation (1) gives the payoff functions of attacker and defender during time period t.(1)$$\begin{array}{c} \left\{\begin{array}{l} U_{D}=(1-\lambda (t))(B-2md)\\ U_{A}=\lambda (t)B+e-2mb\lambda (t) \end{array}\right.\\ s.t.t\leq T \end{array}$$

Clearly, the defender’s payoff function $U_{D}$ is a decreasing function of λ(t), meaning that the defender’s payoff diminishes as their control over public resources is progressively weakened by the ongoing attack. According to our previous research [43], the information entropy metric $F_{No}$ of the redundant executor set $N_{o}$ is also a decreasing function of the attack duration on the redundant executor. Additionally, both $U_{D}$ and $F_{No}$ are continuously differentiable within their respective time domains. Therefore, we assume a linear mapping to conduct an equivalent study of the defense payoff function $U_{D}$ using the information entropy metric $F_{No}$ of the redundant executor set.

**Assumption** **4:**
*There exists a linear mapping φ() with the properties defined in Equation (2), where ${\left(x_{d},x_{a}\right)}^{T}$ and ${\left(y_{d},y_{a}\right)}^{T}$ represent two different linear vectors.*


**Assumption** **5:**
*Each element of the time set T corresponds to a time period of redundant executor scheduling.
*



(2)
$$\left(\varphi (U_{D}),\varphi (U_{A})\right)\to {\left(x_{d},x_{a}\right)}^{T}F_{N_{0}}+{\left(y_{d},y_{a}\right)}^{T}$$


The payoff functions of attacker and defender can be expressed as in Equation (3).(3)$$\begin{array}{c} \left\{\begin{array}{l} U_{D}=F_{No}(t)-2m\varphi (d)\\ U_{A}=F_{No}(\Delta t)+\varphi (e-2mb) \end{array}\right.\\ s.t.\;t\leq T \end{array}$$

According to [43], we can obtain the indicator function expressions Equations (4) and (5) for the information entropy metric and information entropy decay rate in time period t:(4)$$F_{No}(t)=\int_{t}^{T}\log p_{No}(t)f({\hat{s}}_{No})dt$$(5)$$F_{No}(\Delta t)=F_{No}(t_{2})-F_{No}(t_{1})$$

To facilitate subsequent calculations, a discretization of Equations (4) and (5) is performed based on Assumption 5. We define τ to represent the number of samples used for discretizing the information entropy metric and simplify the linear mapping φ() of the constant to the coefficient ε∈[0,1]. Then, the payoff functions can be further expressed as Equation (6).(6)$$\left\{\begin{array}{l} U_{D}=-\sum_{i=1}^{2m}\sum_{k=1}^{\tau }t_{k}(\log p_{i}(t_{k})f({\hat{s}}_{ij}))-2md\cdot \varepsilon \\ U_{A}=-\sum_{i=1}^{2m}\sum_{k=1}^{\tau }t_{c}(\log\frac{p_{i}(t_{k})+p_{i}(\Delta t_{k})}{p_{i}(t_{k})}f({\hat{s}}_{ij}))+(e-2mb)\cdot \varepsilon \end{array}\right.$$

In Equation (6), $t_{ik}$ denotes the duration cycle of redundant executor i in stage k, while $p_{i}(t_{ik})$ represents the probability that redundant executor i is successfully attacked within stage k. The above payoff functions are valid under conditions of perfect information for both the attacker and defender, or perfect information for the defender. Under these conditions, the defender can obtain the optimal solution, while the attacker must adapt his strategy to achieve relatively high payoff.

In the case of memorization-based attacks, the information obtained by attackers and defenders is asymmetric. That is, the attacker possesses all the defender’s vulnerability information, whereas the defender cannot obtain complete information about the attacker. Due to the incomplete information, it is hard, if not impossible, to determine the exact scheduling timing of redundant executors in the DDHR architecture. It is necessary to find an approximate function $T^{\prime }$ for the attacking time to estimate scheduling time T in order to derive the equilibrium optimal solution.

### 3.2. Method for Estimating Scheduling Time T

This section first describes attacking strategy and executor scheduling strategy. Then we present how to estimate scheduling time through estimating the attacking time.

#### 3.2.1. Attacking Strategy

We use a DDHR architecture with a redundancy of 6 to illustrate the attacking strategy. Figure 4 describes the DDHR architecture with redundancy of 6.

Among them, the lower-layers redundant executors $N_{L1},N_{L2}$, and $N_{L3}$ can access each other and can also access the access agent separately. The upper-layer redundant executor $N_{U1},N_{U2}$, and $N_{U3}$ only open secure access links that can be applied for to $N_{L1},N_{L2}$, and $N_{L3}$, respectively, while hiding them from the access agent and other devices.

Then, based on this, we construct a directed weighted attack graph network model $G_{DDHR}=(V,E,C)$ for this DDHR architecture. Here, $V=\{v_{1},v_{2}\ldots v_{6}\}$ represents the set of redundant executor nodes, $E=\{e_{12}\ldots e_{ij}\ldots e_{36}\}$ represents the set of weights of the connecting edges between redundant executors, and *e_ij_* is used to represent the connection weight between redundant executors. In the DDHR architecture, it is equivalent to the similarity metric *s_ij_*. $C=\{c_{1}^{k1}\ldots c_{6}^{k6}\}$ represents the attack status of each node, where k∈[1,τ]. It is illustrated in Figure 5.

**Assumption** **6.**
*In the DDHR architecture, the success of an attack on each redundant executor and the successful detection of attack information are only related to the edge weight indicator $e_{ij}$ and the edge connectivity duration.*


When launching an attack, the attacker can choose any lower-layer node $d_{i}$ to initiate the attack, then repeatedly perform actions such as probing and intrusion along the edges $e_{ij}$ connected to $d_{i}$, ultimately achieving an effective attack by implanting and maintaining the attack payload. In terms of attack strategies, unlike other dynamic redundancy architectures such as MTD and deception defense, due to the specific decision-making mechanism of the DDHR architecture, attackers do not need to target specific objectives. Instead, they adopt the Greedy Exploration strategy [29], which involves selecting the attack strategy with the lowest cost to achieve the highest return within a fixed time frame as the optimal approach.

The attacking strategy for an attacker is detailed as follows:Step 1: Randomly select a node from the lower-layer redundant executor node set $\left\{v_{1},v_{2},v_{3}\right\}$ to conduct attack.Step 2: After the node attack is successful, select the reachable neighboring nodes of the node to conduct detection and obtain the edge weight information $\left\{e_{ij},e_{ik},e_{il}\right\}$.Step 3: Conduct attacks on the target node using edges sorted in ascending order of weights, repeating Step 2 until the attack fails.Step 4: If all attack nodes are successfully attacked, stop the attack.

#### 3.2.2. Heterogeneous Redundant Executor Scheduling Strategy

Under these attack conditions, we propose an anchor-assisting heterogeneous redundant executor scheduling strategy for DDHR attack graph networks, detailed as follows.

We select any node with the largest number of reachable edges among the lower-layer redundant executor nodes as the lower-layer anchor setting position $M_{0}$. (2) Select a redundancy executor from the redundancy resource pool that has the lowest similarity to the reachable edge node and place it at the anchor point. (3) Select an upper-layer redundant executor node that is not connected to the lower-layer node in (1) as the upper-layer anchor point location $M_{1}$. (4) Select a redundant executor from the redundant resource pool with the lowest similarity to the node with the reachable edges and set it at the anchor point location. Algorithm 1 describes the scheduling steps.
**Algorithm 1 Heterogeneous Redundant Executor Scheduling**INPUT: Redundancy resource pool $N_{o}$; redundancy n; redundant resource pool similarity matrix Smatix[*N_o_*]_nxn_; total redundancy of redundant executor in operation r; upper-layer redundant executor set redundancy $m_{up}$; lower-layer redundant executor set redundancy $m_{low}$. The upper-layer execution redundancy executor set is $V_{up}$, the lower-layer execution redundancy executor set is $V_{low}$, and the num() function indicates the current set count.
OUTPUT: Total set of redundant executors *Vr*.
while *i* in $N_{o}$ do: // Initialization scheduling
  1.while *i* in $N_{o}$ do: // Initialization scheduling  2.   If num(*V_up_*) ≤ *m_up_* then  3.     *V_up_* ← *i*  4.   else if num(*V_low_*) ≤ *m_low_*  5.     V_low_ ← i  6.   else  7.     break  8.   endif  9.   Random(*i* in $N_{o}$)// Randomly select redundant executors from *N_0_*  10.endwhile  11.s_asc[*n(n + 1)/2*] ← Select the upper triangular part of Smatix[*N_o_*]_nxn_ and sort it in ascending order  12.// Lower redundant executor anchor point settings  13.for *j* in *V_low_* do:  14.   e_max = 0  15.   for k[*id1,id2*] in s_asc[*n(n + 1)/2*] do:  16.     if (id1 == V_up_(j)) or (id2 == V_low_(j)) then  17.        if e_max < num(k[*id*,_]) then  18.          e_max = num(k[*id*,_])  19.          *V_low_*(*j*) ← *N_o_*(k[*id*])  20.        endif  21.     endif  22.   endfor  23.endfor  24.// Upper redundant executor anchor point settings  25.for *j* in *V_up_* do:  26.   e_max = 0  27.   for k[*id1,id2*] in s_asc[*n(n + 1)/2*] do:  28.      if *V_up_(j)* ≠ *V_low_(j)* then  29.        if (id1 == V_up_(j)) or (id2 == V_up_(j)) then  30.          if e_max < num(k[*id*, _]) then  31.            e_max = num(k[*id*, _])  32.            *V_low_(j) ← N_o_*(k[*id*])  33.          endif  34.        endif  35.     endif  36.   endfor  37.endfor  38.Vr ← V_up_ ∩ V_low_  39.output *Vr*

#### 3.2.3. Estimation of Redundant Executor Scheduling Time

Under the conditions of Assumption 6, the difficulty and time required for an attacker to target anchor heterogeneous nodes are significantly higher than those for non-anchor heterogeneous nodes at the same layer. Therefore, by implementing security monitoring procedures such as increased traffic and system process monitoring on anchor heterogeneous nodes, it becomes easier to detect information related to abnormal attacks.

On the other hand, the links between the lower-layer redundant executor nodes $\left\{v_{1},v_{2},v_{3}\right\}$ and the upper-layer redundant executor nodes $\left\{v_{4},v_{5},v_{6}\right\}$ are subject to the security link time constraints imposed by the DDHR architecture scheduling gateway. As a result, the difficulty and time required to attack the upper-layer redundant executor nodes are also significantly higher than those for redundant executor nodes at the same layer, assuming we can achieve the following conditions through redundant executor configuration:

**Assumption** **7:**
*The attacker spends more time attacking upper-layer redundant entities than same-layer anchor redundant executors and more time attacking same-layer anchor redundant executors than other same-layer non-anchor redundant executors.*


**Assumption** **8:**
*The time when the lower-layer anchor $M_{0}$ is first detected to be under attack is $time_{m0}$, and the time when the upper-layer anchor $M_{1}$ is first detected to be under attack is $time_{m1}$.*


**Assumption** **9:**
*The attacker starts the attack from node $v_{1}$, and the attack follows the distribution function ρ(t) on time T.*


Therefore, the attack route for $G_{DDHR}$ can be determined as path 1: $v_{1}\to v_{3}\to v_{2}\to v_{4}/v_{6}\to v_{5}\to v_{6}/v_{4}$ or path 2: $v_{1}\to v_{2}\to v_{3}\to v_{4}/v_{5}\to v_{6}\to v_{5}/v_{4}$.

At the same time, regardless of whether an attack is detected, during every time interval μ, the lower-layer or upper-layer redundant executor set is scheduled once. Taking path 1 as an example, the time when node $v_{1}$ is attacked is given by Equation (7).(7)$${t^{\prime }}_{v1}=\left\{\begin{array}{l} \rho (time_{m0}/f{\left(s_{1M0}\right)}^{-1}),{t^{\prime }}_{v1}<\mu \\ u,{t^{\prime }}_{v1}\geq \mu \end{array}\right.$$

The time when the *v_3_* node was attacked is given by Equation (8).(8)$${t^{\prime }}_{v3}=\left\{\begin{array}{l} \rho (time_{m0}/f{\left(s_{3M0}\right)}^{-1}),{t^{\prime }}_{v3}<\mu \\ u,{t^{\prime }}_{v3}\geq \mu \end{array}\right.$$

The time when the $\nu_{2}$*v_2_* node was attacked is given by Equation (9).(9)$${t^{\prime }}_{v2}=time_{m0}-{t^{\prime }}_{v3}-t\prime_{v1}$$

The time when the $v_{4/}v_{6}$ node was attacked is given by Equation (10).(10)$${t^{\prime }}_{v4/6}=\left\{\begin{array}{l} \rho ((time_{m1}-time_{m0})/f{\left(s_{4/6M1}\right)}^{-1}),{t^{\prime }}_{v4/6}<\mu \\ u,{t^{\prime }}_{v4/6}\geq \mu \end{array}\right.$$

The time when the $v_{5}$ node was attacked is given by Equation (11).(11)$${t^{\prime }}_{v5}=\left\{\begin{array}{l} \rho ((time_{m1}-time_{m0})/f{\left(s_{5M1}\right)}^{-1}),{t^{\prime }}_{v5}<\mu \\ u,{t^{\prime }}_{v5}\geq \mu \end{array}\right.$$

The time when the *v_6/_v_4_* node was attacked is given by Equation (12).(12)$${t^{\prime }}_{v6/4}=time_{m1}-{t^{\prime }}_{v5}-{t^{\prime }}_{v4/6}$$

In Equations (7)–(12), f( ) represents the function that describes the impact of similarity between two redundant executors on redundant executor scheduling time. The approximate estimate of redundant executor scheduling time T is $T^{\prime }=\{{t^{\prime }}_{v1},{t^{\prime }}_{v2},{t^{\prime }}_{v3},t_{v4}^{\prime },t_{v5}^{\prime },t_{v6}^{\prime }\}$. Redundant executor scheduling time updates can be dynamically adjusted based on the attack time after each attack on the anchor redundant executor. Algorithm 2 describes the attack time estimation steps.
**Algorithm 2: Attack Time Estimation Method**INPUT: The lower anchor point $M_{0}$ was first detected to be attacked at time $time_{m0}$, and the upper anchor point $M_{1}$ was first detected to be attacked at time $time_{m1}$; the redundancy degree of the redundancy executors set is 2m; the fixed scheduling time threshold is μ; the distribution function ρ(t) of the attack on time *T*.
OUTPUT: Set *T′* of approximate estimates of redundant executor scheduling time *T*.
  1.*T′* ← Φ  2.for *i* in *T′* do  3.   if *T′*! = Φ then  4.     if *T*′[*i*] == μ then  5.        *J* = *j* + 1  6.       endif  7.   else  8.    *T′* ← Equations (7)–(12) calculates attack time for each node redundant executor.  9.   endif  10.endfor  11.if *j* ≥ 2*m* − 2 then  12.   Update ρ(t) and re-execute Algorithm 2.  13.else  14.   output *T′*  15.endif


### 3.3. Optimal Defender’s Payoff Model Under Approximately Complete Information

Based on the algorithms in Section 3.2, we can transform incomplete information conditions into approximate complete information conditions and then estimate the redundant executor scheduling time $T^{\prime }$. Then we can consider the constraints in Equation (6) to ensure that the FlipIt-game is the optimal payoff model.

(1)As the initiator of the game, the attacker’s payoff cannot be negative to ensure the existence of an equilibrium solution, i.e., it must satisfy Equation (13).


(13)
$$-\sum_{i=1}^{2m}\sum_{k=1}^{\tau }t_{ik}^{\prime }(\log\frac{p_{i}(t_{ik}^{\prime })+p_{i}(\Delta t_{ik}^{\prime })}{p_{i}(t_{ik}^{\prime })}f({\hat{s}}_{ij}))+(e-2mb)\cdot \varepsilon \geq 0$$


(2)The defender’s payoff must not fall below the minimum payoff required for the DDHR architecture to take effect, meaning the defender’s payoff $U_{D}$ must be larger than the sum of the defense payoff of any (2m+1)/2 redundant executors, satisfying Equation (14).


(14)
$$-\sum_{i=1}^{\left[(2m+1)/2\right]}\sum_{k=1}^{\tau }t_{ik}^{\prime }(\log p_{i}(t_{ik}^{\prime })f({\hat{s}}_{ij}))\geq 0$$


(3)We set the number of heterogenous redundant anchor points to X. The entropy of the $\left\{M_{0}\ldots M_{X}\right\}$ heterogeneous anchor points must be larger than zero to ensure that the scheduling time $T^{\prime }$ is accurately estimated during each scheduling, thereby satisfying Equation (15).


(15)
$$\left\{\begin{array}{l} -\sum_{k=1}^{\tau }t_{ik}^{\prime }(\log p_{M0}(t_{ik}^{\prime })f({\hat{s}}_{M0j}))>0\\ \vdots \\ -\sum_{k=1}^{\tau }t_{ik}^{\prime }(\log p_{Mx}(t_{ik}^{\prime })f({\hat{s}}_{Mxj}))>0 \end{array}\right.$$


In Equations (13)–(15), $t_{ik}^{\prime }$ denotes the duration of redundancy executor i at stage k, estimated through the heterogenous redundant executor anchor point configuration. Furthermore, the defender’s optimal equilibrium payoff can be expressed by Equation (16).(16)$$\begin{array}{l} \max R_{D}=\max(-\sum_{i=1}^{2m}\sum_{k=1}^{\tau }\left(t_{ik}^{\prime }\log p_{i}(t_{ik}^{\prime })f({\hat{s}}_{ij})\right)-2md\cdot \varepsilon )\\ s.t.\left\{\begin{array}{l} -\sum_{i=1}^{2m}\sum_{k=1}^{\tau }t_{ik}^{\prime }(\log\frac{p_{i}(t_{ik}^{\prime })+p_{i}(\Delta t_{ik}^{\prime })}{p_{i}(t_{ik}^{\prime })}f({\hat{s}}_{ij}))+(e-2mb)\cdot \varepsilon \geq 0\\ -\sum_{i=1}^{\left[(2m+1)/2\right]}\sum_{k=1}^{\tau }t_{ik}^{\prime }(\log p_{i}(t_{ik}^{\prime })f({\hat{s}}_{ij}))\geq 0\\ -\sum_{k=1}^{\tau }t_{ik}^{\prime }(\log p_{M0}(t_{ik}^{\prime })f({\hat{s}}_{M0j}))>0\\ \vdots \\ -\sum_{k=1}^{\tau }t_{ik}^{\prime }(\log p_{Mx}(t_{ik}^{\prime })f({\hat{s}}_{Mxj}))>0 \end{array}\right. \end{array}$$

## 4. PPO-Based Approach for Optimizing DDHR Redundant Executor Scheduling

This section first presents the DDHR scheduling framework where a PPO-based approach is used for scheduling redundant executors. Then Section 4.2 describes the MDP for Equation (16), based on which the PPO-based approach is presented in Section 4.3.

### 4.1. DDHR Scheduling Framework

Figure 6 illustrates the FlipIt-DDHR scheduling framework based on the PPO-HE approach. We detail the framework as follows.

(1)State Collection Module: Real-time connection to the redundant executor scheduling strategy gateway, primarily responsible for adjusting scheduling strategies, receiving information entropy changes in redundant executors under the current strategy, and calculating the current strategy stage payoff function, then storing the results in the experience pool.(2)Strategy Network Module: Real-time connection to the state collection module, receiving state changes and outputting corresponding actions. Simultaneously, connected to the optimizer to update strategy parameter θ.(3)Evaluation Network Module: Real-time connection to the state collection module, receiving complete trajectories $tr=\{(s_{k},a_{k},r_{k},s_{k+1})\ldots (s_{\tau },a_{\tau },r_{\tau },s_{\tau +1})\}$ from multiple stages. Then, it calculates the advantage function to update the strategy network. Simultaneously, it connects to the optimizer and updates the evaluation network parameters ϕ.(4)Optimizer Module: Real-time connection to the strategy network module and evaluation network module, updating strategy parameters θ and evaluation network parameters ϕ.

The approach workflow is shown in Figure 7.

Firstly, the income agent publishes the task to redundant executors $N_{1}$ to $N_{2m}$ (which contain tasks that have been attacked and executed incorrectly or failed). Secondly, the redundant executors calculate the results separately and send the results to the arbiter, which outputs the final execution results and the status of each redundant executor to the status monitoring module. Thirdly, the state monitoring module calculates the information entropy and anchor redundant executor similarity function, and on one hand performs redundant executor scheduling action A_t(θ_old) based on the policy θ_old published by the policy network module, and on the other hand sends the complete trajectory <S_t,A_t,r_t,S_t+1> to the evaluation network. Fourthly, the evaluation network calculates the advantage function $\hat{A}\_t$ and the loss function of the critic network L_V(φ) and sends L_V(φ) and V(φ) to the optimizer. Meanwhile, the policy network calculates the loss function of the policy network L_clip(θ) and sends L_clip and $\pi_{\theta }$ to the optimizer. Finally, the optimizer calculates the total loss L, the gradient of θ, and the gradient of φ, and updates the gradients to the policy and critic networks, completing a full optimization cycle.

### 4.2. Modeling Markov Decision Processes for FlipIt-DDHR Scheduling Decision Problem

We construct an MDP model, which is $<S_{SDP},A_{SDP},R_{SDP}>$, based on the DDHR architecture information entropy metric where $S_{SDP}$ is the state space, $A_{SDP}$ is the action space, and $R_{SDP}$ is the reward function.

#### 4.2.1. State Space

According to the attack graph network model $G_{DDHR}=(V,E,C)$ in Section 3.2, we extend the nodes to 2m (with redundant executor nodes of m in both upper and lower layers). At stage k, k∈τ, and the redundant executor state set C can be expressed as $C=\{c_{1}^{k1}\ldots c_{2m}^{k2m}\}$. For any redundant executor i, the state $c_{i}^{ki}$ satisfies $k_{i}\in [1,\tau ]$. Additionally, we consider the heterogeneity of x heterogeneous anchor points. After changes in the connectivity of node scheduling, these anchor points need to be reconfigured. Therefore, the state space should also include the similarity states of the x heterogeneous redundant executor anchor points. Thus, we represent the MDP model state space $S_{SDP}$ as in Equation (17):(17)$$S_{SDP}=\{F_{V}(t_{k}),F_{V}(\Delta t_{k}),f_{Mx}(s_{ij}|e_{i})\}$$

Among them, $F_{V}(t_{k})$ represents the sum of the redundant executor information entropy indices of the redundant executors set V at time k in state $t_{k}$. $F_{V}(\Delta t_{k})$ represents the change in information entropy metric of the redundant executors set V from state k−1 to state k over time interval $t_{k}$. $f(s_{ij}|e_{ij})$ represents the total similarity weight function of the redundant executor i interconnection path $e_{i}$.

#### 4.2.2. Action Space

According to the research results in [43], the scheduling strategy of the DDHR architecture includes two types: weight adjustment and redundant executor replacement. At the same time, based on the state, the setting action of the heterogeneous redundant executor anchor point is added. Therefore, we can define the action space as in Equation (18):(18)$$A_{SDP}=\{\omega (c_{i}^{ki}),\sigma (c_{i}^{ki}),\kappa (s_{M_{X}})\}$$

Here, $\omega (c_{i}^{ki})$ is the weight adjustment function for the redundant executor arbitration, whose range is determined by the redundant executor state weight function in [43]. $\sigma (c_{i}^{ki})$ is the redundant executor scheduling function, and its range is determined by the total number of schemes included in the redundant executor resource pool. $\kappa (s_{M_{X}})$ is the setting action function for the similarity of heterogeneous redundant executor anchor points.

#### 4.2.3. Reward Function

The total reward function $R_{SDP}(s_{SDP},a_{SDP})$ of the system can be jointly expressed by the defender’s payoff function $R_{D}$ from Section 3.3 and the total effective reward $R_{sur}(t_{ik}^{\prime })$ of the DDHR architecture. Without considering the costs of attack and defense, the reward function can be represented by Equation (19). α and β are normalization parameters.(19)$$\begin{array}{l} R_{SDP}(s_{SDP},a_{SDP})=[\max(-\sum_{i=1}^{2m}\sum_{k=1}^{\tau }\left(t_{ik}^{\prime }\log p_{i}(t_{ik}^{\prime })f({\hat{s}}_{ij})\right))]\cdot \alpha +R_{sur}(t_{ik}^{\prime })\cdot \beta \\ s.t.\left\{\begin{array}{l} -\sum_{i=1}^{2m}\sum_{k=1}^{\tau }t_{ik}^{\prime }(\log\frac{p_{i}(t_{ik}^{\prime })+p_{i}(\Delta t_{ik}^{\prime })}{p_{i}(t_{ik}^{\prime })}f({\hat{s}}_{ij}))\geq -e\varepsilon \\ -\sum_{i=1}^{\left[(2m+1)/2\right]}\sum_{k=1}^{\tau }t_{ik}^{\prime }(\log p_{i}(t_{ik}^{\prime })f({\hat{s}}_{ij}))\geq 0\\ -\sum_{k=1}^{\tau }t_{ik}^{\prime }(\log p_{M0}(t_{ik}^{\prime })f({\hat{s}}_{M0j}))>0\\ \vdots \\ -\sum_{k=1}^{\tau }t_{ik}^{\prime }(\log p_{Mx}({t^{\prime }}_{ik})f({\hat{s}}_{Mxj}))>0\\ \alpha +\beta =1 \end{array}\right. \end{array}$$

### 4.3. PPO_EH Approach

The objective function of PPO_EH can be expressed as Equation (20):(20)$$J^{\mathrm{ppo}}=\underset{\theta }{\max}\underset{st,at∼\pi \theta old}{E}[L(s_{t},a_{t},\theta_{old},\theta )]$$

We need to continuously update the parameters θ and ϕ of the policy network and critic network to achieve the optimal goal. Algorithm 3 describes the approach, detailed in the following.


**Algorithm 3 PPO_EH Approach**
INPUT: According to the attacker strategy and defender scheduling strategy environment in Section 3.2, set hyperparameters ξ and γ.
OUTPUT: Updated network parameters θ, ϕ.
  1.According to Algorithms 1 and 2, initialize the environmental information entropy metric $F_{V}(t_{k})$, the initial state $s_{0i}=F_{i}(t_{0})\cdot g(f_{Mx}(s_{ij}|e_{ij}))$, and initialize the policy network parameters $\theta_{0}$  2.Initialize the value network parameter $\phi_{0}$  3.for epoch in range(EPOCHS) do  4.   Experience pool *D* ← Ø  5.   for *i = 1 I* do  6.     Initialize trajectory *tr* ← Ø  7.     for *k* = 1 τ do  8.         Obtain the current redundant executor set status $s_{k}=\sum_{i=1}^{2m}F_{i}(t_{k})\cdot g(f_{Mx}({\hat{s}}_{ij}))$  9.         According to the old strategy network $\theta_{old}$, obtain the current redundant executor scheduling action $a_{k}$  10.       Calculate $r_{k}$ using Equation (19).  11.       Calculate the next stage state $s_{k+1}$  12.       *tr* ← $tr\cup (s_{k},a_{k},r_{k},s_{k+1})$  13.     endfor  14.     D←D∪tr  15.   endfor  16.Update strategy network parameters θ according to Equation (23).  17.Update strategy network parameters ϕ according to Equation (25).  18.Update old network parameters $\theta_{old}\leftarrow \theta$  19.endfor


#### 4.3.1. Initialize the Information Entropy Environment

The specific steps for initializing the information entropy environment are as follows: First, calculate the information entropy metric $F_{V}(t_{k})$ for the redundant executor and redundant resource pool at the anchor point location based on Algorithms 1 and 2 and Equation (19). Then, for any redundant executor i, we can express its initial state $s_{0}$ using the information entropy metric and the anchor point position redundant executor similarity weighting function, i.e., $s_{0i}=\{F_{i}(t_{0}),f_{Mx}(s_{ij}|e_{ij})\}$. Finally, according to the formula for calculating the information entropy index in Equation (19), for redundant executors at non-anchor positions, $s_{0i}=F_{i}(t_{0})\cdot 1$; for redundant executors at anchor positions, $s_{0i}=F_{i}(t_{0})\cdot g(f_{Mx}({\hat{s}}_{ij}))$. Among these, g() represents the ratio function of anchor position redundancy similarity to average similarity. This ultimately yields the computable state value $s_{0i}$.

#### 4.3.2. Motion Sampling

Initialize the policy $\pi_{\theta }$ using the random scheduling and entropy weight variation approach described in Reference [43]. Use $\pi_{\theta }$ for redundant executor scheduling and collect sample traces $tr=\{s_{k},a_{k},r_{k},s_{k+1},\ldots \}$ for each stage through the collection module and store them in the experience pool.

#### 4.3.3. Calculate the Advantage Function

The value function $V(s_{k})$ for state *k* is computed based on the latest state of each trajectory in the experience pool. It can be recursively calculated by the critic network using the state value $s_{k}$ of each trajectory—specifically, the sum of the redundant executor information entropy $\sum_{i=1}^{2m}F_{i}(t_{k})$ for state *k* across all trajectories. Then, the TD residual is calculated by using the GAE method, as expressed in Equation (21).


(21)
$$\delta_{k}=r_{k}+\gamma V(s_{k+1})-V(s_{k})$$


Then, calculate the advantage function ${\hat{A}}_{k}$ recursively in reverse order, as expressed by Equation (22).(22)$${\hat{A}}_{k}=\delta_{k}+\gamma \lambda_{k+1}$$

#### 4.3.4. Update the Policy Network

To ensure the stability of the update, we use the PPO-clip method for policy network updates. The loss function is in Equation (23).(23)$$L^{clip}(s_{k},a_{k},\theta_{old},\theta )=E[\min(r_{k}(\theta ){\hat{A}}_{k},clip(r_{k}(\theta ),1-\xi ,1+\xi ){\hat{A}}_{k})]$$

Here, ξ is the truncation constant, and $r_{k}(\theta )=\frac{\pi_{\theta }(a_{k}|s_{k})}{\pi_{\theta old}(a_{k}|s_{k})}$ is the ratio of the probabilities of the new and old strategies. The chip function is given by Equation (24).(24)$$clip(r_{k}(\theta ),1-\xi ,1+\xi )=\left\{\begin{array}{l} 1+\xi ,r_{k}(\theta )\geq 1+\xi \\ 1-\xi ,r_{k}(\theta )\leq 1+\xi \\ r_{k}(\theta ) \end{array}\right.$$

#### 4.3.5. Updating the Critic Network

The loss function of the critic network can be expressed by Equation (25).(25)$$L(\phi )=E[{\left(\sum_{i=k}^{\tau }\gamma^{i-k}r_{i}-V_{\phi }(s_{k})\right)}^{2}]$$

## 5. Experiment Evaluation

This section evaluates the performance metrics of the PPO-EH scheduling algorithm under memorization-based attack within an internal cloud network environment using the FlipIt-DDHR architecture. It also compares and analyzes this algorithm against other redundant scheduling algorithms. The experimental environment is shown in Table 3.

### 5.1. Initialize Environment Settings

#### 5.1.1. Initialization of Information Entropy of Redundant Executor Set

According to Equation (19), this initialization comprises four components: initialization of the redundancy degree in the redundant resource pool, initialization of the similarity weight for redundant executors, initialization of heterogeneous anchor point redundant executors, and initialization of the scheduling time.

(1)Initialization of Redundancy Resource Pool

Considering both the economic feasibility of building the redundancy resource pool and the dynamic scheduling capabilities, this paper selects a total redundancy n = 12 for the pool. The total redundancy 2m for the online redundancy sets are set to 6 and 8, respectively. Two sets of experiments are conducted when the upper-layer redundancy set and lower-layer redundancy set contain 3 and 4 elements, respectively. The information entropy of the redundant executor after an attack does not increase due to scheduling replacement. The arbitration strategy adopts a majority consensus approach, requiring that the number of redundant executors with consistent results be no less than (2m+1)/2.

(2)Initialization of Redundant Executor Similarity Weight $f({\hat{s}}_{ij})$

The similarity weight parameters in this paper are randomly generated from the β- distribution with parameters (5, 15). The similarity between redundant executors i and j is represented as a 12 × 12 matrix, as shown in Equation (26).


(26)
$$\left(\begin{array}{cccccccccccc} 1 & 0.415 & 0.259 & 0.417 & 0.306 & 0.485 & 0.446 & 0.320 & 0.335 & 0.201 & 0.260 & 0.347\\ 0.415 & 1 & 0.232 & 0.294 & 0.354 & 0.490 & 0.411 & 0.347 & 0.169 & 0.459 & 0.406 & 0.323\\ 0.259 & 0.232 & 1 & 0.253 & 0.254 & 0.205 & 0.233 & 0.455 & 0.211 & 0.384 & 0.375 & 0.209\\ 0.417 & 0.294 & 0.253 & 1 & 0.403 & 0.400 & 0.157 & 0.415 & 0.408 & 0.215 & 0.261 & 0.261\\ 0.306 & 0.354 & 0.254 & 0.403 & 1 & 0.421 & 0.324 & 0.148 & 0.328 & 0.339 & 0.251 & 0.250\\ 0.485 & 0.490 & 0.205 & 0.400 & 0.421 & 1 & 0.405 & 0.152 & 0.494 & 0.394 & 0.289 & 0.332\\ 0.446 & 0.411 & 0.233 & 0.157 & 0.324 & 0.405 & 1 & 0.484 & 0.235 & 0.194 & 0.335 & 0.464\\ 0.320 & 0.347 & 0.455 & 0.415 & 0.148 & 0.152 & 0.484 & 1 & 0.495 & 0.141 & 0.347 & 0.254\\ 0.336 & 0.169 & 0.211 & 0.408 & 0.328 & 0.494 & 0.235 & 0.495 & 1 & 0.292 & 0.275 & 0.338\\ 0.201 & 0.459 & 0.384 & 0.215 & 0.339 & 0.394 & 0.194 & 0.141 & 0.292 & 1 & 0.339 & 0.298\\ 0.260 & 0.406 & 0.375 & 0.261 & 0.251 & 0.289 & 0.335 & 0.347 & 0.275 & 0.339 & 1 & 0.426\\ 0.348 & 0.323 & 0.209 & 0.261 & 0.251 & 0.332 & 0.464 & 0.254 & 0.338 & 0.298 & 0.426 & 1 \end{array}\right)$$


(3)Initialization of Heterogeneous Redundant executor Anchor points

We configure the network with redundancy of 6 and 8, where lower-layer redundant executors are interconnected, while upper-layer redundant executors connect only to a single lower-layer redundancy executor. Additionally, under both redundancy conditions, the lower-layer redundant executor $v_{2}$ and upper-layer redundant executor $v_{6}$ are designated as anchor redundant executors, as illustrated in Figure 8.

(4)Initialization of Redundant Executor Scheduling Time

To facilitate the calculation of information entropy environment values and experimental result comparisons, the attack time setting in this paper adopts an attack cycle function that satisfies a uniform distribution over the estimated total occupation time $T^{\prime }$. This is achieved by satisfying either Equations (27) or (28).(27)$$\rho (time_{m0}/f{\left(s_{iM0}\right)}^{-1})=time_{m0}/f{\left(s_{iM0}\right)}^{-1}$$(28)$$\rho (time_{m1}/f{\left(s_{iM1}\right)}^{-1})=time_{m1}/f{\left(s_{iM1}\right)}^{-1}$$

The attack time for an anchor redundancy executor is 1.25 times the average attack time of its connected anchor redundant executors. Regarding attack strategies, this paper assumes the attacker employs a greedy attack strategy [29]. Attack paths consistently originate from the lower-layer redundant executor, with attack frequency matching the average attack time, and each attack targets only one redundant executor.

#### 5.1.2. Initializing Experimental Parameters

The parameter settings for the approach training and simulation experiment phases are shown in Table 4. The values of the first parameters are set according to the literature related to PPO algorithm. The values of the last four parameters are set based on our experiments.

### 5.2. Training Experiment Analysis

In this phase, we first conducted comparative experiments by adjusting key hyperparameters that influence algorithm performance, including learning rate, discount factor, and clip ratio. Subsequently, we selected the optimal convergence scheme among them for comparative experiments against the traditional DHR architecture under the information entropy environment using the PPO approach (PPO_E), as well as the PPO-based self-evolving moving target defense strategy under information entropy (PPO) [34].

#### 5.2.1. Average Rewards for PPO-EH Approach Under Different Learning Rate

The learning rate parameter is the most critical factor affecting algorithm convergence. Therefore, this paper first conducts experiments with learning rates ranging from 10^−3^ to 10^−5^. For redundancy 2m = 6 and 8, respectively, four sets of experiments were conducted at 1 × 10^−3^, 5 × 10^−3^, 1 × 10^−4^, and 5 × 10^−5^ for redundancy of 6, 8. The rewards are shown in Figure 9. The average reward value (AR) is shown in Table 5.

In summary, when redundancy is 6 and 8, the average reward value under a learning rate of 1 × 10^−4^ was higher than under other conditions. Regarding convergence speed, the 5 × 10^−3^ learning rate converged fastest, though all learning rates converged sufficiently around 9000 steps. In terms of stability, the 1 × 10^−4^ learning rate demonstrated superior stability compared to others. Therefore, we selected the 1 × 10^−4^ learning rate for comparison with other approaches.

#### 5.2.2. Average Rewards for PPO-EH Approach Under Different Clip Ratio

Based on the research in [33] and common experience with the PPO algorithm, the clip ratio parameter primarily helps truncate updates with excessive magnitude. It is typically selected within the range [0.1–0.3]. This paper conducted comparative experiments using values of 0.1, 0.2, and 0.3 under a learning rate of 1 × 10^−4^, with redundancy of 6 and 8. The experimental results are shown in Figure 10.

The average reward values under different clip ratios are shown in Table 6:

From the above experiments, it can be observed that a larger clip ratio parameter results in faster learning speed and higher average rewards at the same step, but with slightly weaker stability. Conversely, a smaller clip ratio leads to slower learning speed and lower average rewards at the same step, but with slightly stronger stability. In this model, the difference between these two scenarios is relatively small. When clip ratio is set to 0.2, the overall stability is optimal for redundancy levels of 6 and 8. Therefore, subsequent experiments all adopt clip ratio = 0.2.

#### 5.2.3. Average Rewards for PPO-EH Approach Under Different Discount Factor

Based on common experience with the PPO algorithm, the discount factor parameter primarily controls the scope of the algorithm’s learning experience, typically selected within the range [0.9–0.999]. This paper conducted comparative experiments using the values 0.9, 0.99, and 0.999 under a learning rate of 1 × 10^−4^, with redundancy levels of 6 and 8. The experimental results are shown in Figure 11:

The average reward values under different discount factor parameters are shown in Table 7:

The experiments above indicate that in this model, the overall reward value differences are relatively small. When the discount factor is set to 0.99, the average reward value is highest at redundancy levels of 6 and 8, while also exhibiting the best overall stability. Therefore, subsequent experiments will all use a discount factor of 0.99.

#### 5.2.4. Average Rewards for PPO-EH Approach Under Different Weight Parameters α,β of the Reward Function

According to Formula (19), a and b are the normalization parameters for the control information entropy loss reward and the defensive survival reward (i.e., the scheduling cycle reward). In this model, the information entropy loss reward serves as the primary reward, while the survival reward—a linearly increasing function of resource occupation time *t*—acts as a passive reward. Therefore, we use the survival reward weight parameter *beta* as the primary variable, conducting comparative experiments with values of 0, 0.2, 0.4, 0.6, 0.8, and 1.

It should be noted that to ensure agent training speed, when beta is set within [0–0.8] by default, it is set to 0 for the first 1000 steps and adjusted according to the above ratios thereafter. The experimental results are shown in Figure 12.

The average reward values under different beta values are shown in Table 8:

The experiments above show that the average reward is maximized when *beta* is set to 0.4, corresponding to {α=0.6, β=0.4}. Furthermore, when β<0.4, the agent’s reward increases as the survival reward weight rises. Conversely, when β>0.4, the agent receives diminishing entropy reduction rewards as the survival reward weight increases, leading to a reduction in the learned scheduling strategy. When β=1, the agent essentially ceases scheduling actions, relying solely on the survival reward. Therefore, subsequent experiments in this paper are conducted with {α=0.6, β=0.4}.

#### 5.2.5. Comparison of Different Approaches

At redundancy of 6 and 8, we select two approaches to compare with our PPO_EH, namely, PPO_E and PPO approaches. We detail them as follows.

**PPO_E Approach**: Adjusts the information entropy environment of the PPO_EH approach by removing anchor point settings and the linkage constraints between upper and lower layers, thereby transforming it into a DHR architecture without connectivity restrictions. Consequently, the conditions in Equation (19) can be modified to Equation (29):(29)$$s.t.\left\{\begin{array}{l} -\sum_{i=1}^{2m}\sum_{k=1}^{\tau }{t^{\prime }}_{ik}(\log\frac{p_{i}({t^{\prime }}_{ik})+p_{i}(\Delta {t^{\prime }}_{ik})}{p_{i}({t^{\prime }}_{ik})}f(s_{ij}))\geq \text{-}e\varepsilon \\ -\sum_{i=1}^{\left[(2m+1)/2\right]}\sum_{k=1}^{\tau }{t^{\prime }}_{ik}(\log p_{i}({t^{\prime }}_{ik})f(s_{ij}))\geq 0 \end{array}\right.$$

**PPO Approach**: Assuming attack information in the DRL-SEMTD approach is fully identifiable, under identical environmental conditions as the PPO_EH approach, remove the constraint on the information entropy decay threshold. Additionally, impose a requirement that any movement strategy must ensure the change in information entropy metrics from stage k−1 to stage k is less than zero, thereby guaranteeing convergence and stability during training. Consequently, the conditions in Equation (19) can be modified to Equation (30):(30)$$s.t.\left\{\begin{array}{l} -\sum_{i=1}^{2m}\sum_{k=1}^{\tau }{t^{\prime }}_{ik}(\log\frac{p_{i}({t^{\prime }}_{ik})+p_{i}(\Delta {t^{\prime }}_{ik})}{p_{i}({t^{\prime }}_{ik})}f({\hat{s}}_{ij}))\geq \text{-}e\varepsilon \\ \sum_{i=1}^{\left[(2m+1)/2\right]}(k{t^{\prime }}_{ik}(\log p_{i}({t^{\prime }}_{ik})f({\hat{s}}_{ij})))<0 \end{array}\right.$$

Under the aforementioned adjustment conditions, the training results comparison is shown in Figure 13.

The average reward value (AR) for each approach is shown in Table 9.

In summary, as the online redundancy executor increases, the average reward values obtained by each approach gradually rise, indicating that the defender’s potential gains progressively increase and the defensive strategies become more effective. Regarding approach performance, the PPO approach employing the moving target defense strategy yields significantly lower average reward values than both PPO_EH and PPO_E approaches. At redundancy of 6, its values are 0.459 times and 0.439 times those of PPO_EH and PPO_E, respectively. At redundancy of 8, they are 0.484 times and 0.462 times the PPO approach’s values. Additionally, while convergence is achievable, the convergence speed is significantly slower than that of the PPO_EH and PPO_E approaches. This indicates that the PPO approach with a mobile target defense strategy can effectively prevent attacks using a single redundant executor during task execution. However, it offers limited enhancement to the overall defense capability of the architecture and exhibits slower optimization speed for finding optimal defense strategies. The PPO_EH approach yields a smaller average reward than the PPO_E approach, specifically achieving 0.956 times and 0.954 times the reward of the PPO_E approach at redundancy levels of 6 and 8, respectively. Although the rate of reward change during the ascent phase is slightly lower than that of the PPO_E approach, the average reward during the stable phase is marginally higher than that of the PPO_E approach. This indicates that by increasing the number of heterogeneous redundant executor anchor points and imposing connectivity constraints on redundant executors, the overall environmental complexity of the PPO_EH approach is higher than that of the PPO_E approach. After training stabilization, the PPO_EH approach demonstrates stronger resilience against attacks, a finding supported by subsequent simulation experiments.

### 5.3. Analysis of the Capability of PPO-EH Approach

In this phase, we selected the aforementioned three approaches and supplemented them with the traditional random scheduling approach and the REWS approach from [43] for comparison in redundant executor scheduling. The attack information entropy decay rate was set to 0.05 for each iteration to prevent excessive numerical discrepancies among the approaches.

#### 5.3.1. Analysis in Dynamic Experiment

The dynamic performance of redundant executor scheduling approaches serves as a critical metric for evaluating the capabilities of DHR-type architectures. Under conditions of guaranteed heterogeneity, approaches with superior dynamic performance better ensure the reliability of DHR architectures. We conducted 100 independent experiments on the five approaches using a scheduling cycle T, with each cycle duration measured in units of $t_{mean}$. The results are shown in Figure 14.

The average scheduling cycle and experimental variance for different approaches at redundancy levels of 6 and 8 are shown in Table 10.

In summary, regarding dynamic performance, (1) the PPO_EH approach achieves average scheduling cycles of 103.73 $t_{mean}$ and 121.65 $t_{mean}$ for redundancy of 6 and 8, respectively. While slightly lower than the PPO_E approach, it demonstrates superior dynamic performance compared to other approaches—meaning it provides longer effective defense against memorization-based attacks than alternative approaches. (2) The average scheduling cycle of the PPO_EH approach is slightly lower than that of the PPO_E approach, primarily due to constraints imposed by the DDHR architecture and the configuration of the heterogeneous anchor redundancy set on dynamic scheduling. (3) Under constant total redundancy of 12, increasing the online redundant executors set redundancy from 6 to 8 reduces the redundancy scheduling cycle. This indicates that increasing online redundant executors in limited resources diminishes the architecture’s dynamic capability. As demonstrated by the experimental results in Section 5.1(2), increasing the redundancy of the online redundant executors set enhances its resistance to attacks but sacrifices dynamic capability. Therefore, under conditions of incomplete information, the PPO_EH approach dynamically adjusts the number of online redundant executors based on estimated attack intensity (attack timing), offering advantages over other approaches in such scenarios. Regarding stability, the variance of reinforcement learning-based approaches PPO_EH, PPO_E, and PPO is lower than that of stochastic approaches, indicating that reinforcement learning-based redundancy scheduling approaches can effectively enhance the stability of DDHR and DHR architectures.

#### 5.3.2. Analysis of Online Redundant Executors Set Average Scheduling Efficiency

The online redundant executors set average scheduling efficiency is represented by two metrics: total information entropy decay amount per scheduling instance and the information entropy decay rate. Total information entropy decay amount is denoted as $F_{V}(\Delta t_{k})$, while the information entropy decay rate $F_{V}{\left(\Delta t_{k}\right)}_{ratio}$ is the ratio of the reduced information entropy to the average scheduling cycle, as shown in Equation (30).(31)$$F_{V}{\left(\Delta t_{k}\right)}_{ratio}=F_{V}(\Delta t_{k})/Ts_{avg}$$

During each scheduling process, a smaller $F_{V}(\Delta t_{k})$ value indicates that the online redundant executor can protect against memorization-based attacks for a longer duration. While ensuring the effectiveness of the architecture, a smaller $F_{V}{\left(\Delta t_{k}\right)}_{ratio}$ value signifies higher scheduling quality per iteration, meaning that the architecture’s effectiveness is maintained at a lower cost. Similarly, after conducting 100 independent experiments, the results are shown in Figure 15.

The information entropy decay amount and decay rate for different approaches at redundancy of 6 and 8 are shown in Table 11.

In summary, the PPO_EH approach achieves an average information entropy decay amount of only 19.08 and 18.04 at redundancy of 6 and 8, respectively. Excluding the Random approach with the lowest initial entropy metric, this performance falls significantly below other approaches. Moreover, its information entropy decay rates were only 0.16 and 0.17, representing 72.7% and 77.3% of the PPO_E approach (the lowest among other approaches) and 17.2% and 17.5% of the Random approach (the highest among other approaches). This indicates that the PPO_EH approach achieves the highest efficiency in each attack scheduling and provides the longest resistance time against memorization-based attacks under identical initial information entropy conditions.

## 6. Conclusions

This paper aims to resist APT threats by scheduling DDHR when the defender has incomplete information in DDHR architecture. We first propose a DDHR architecture attack–defense payoff model based on the FlipIt game strategy. Building upon this, it introduces an optimization method for scheduling redundant executors in the DDHR architecture using introducing heterogeneity to DDHR redundant executors. This heterogeneity is generated by defining some special points (namely, anchors) for deploying executors to cause diverse connectivity from these anchors to the other executors. Subsequently, the PPO_EH approach is designed to solve the payoff model. Experimental results indicate that the PPO_EH approach achieves the highest average reward return during stable operation.

There exists some limitations with this paper. This paper aims to explore the importance of entropy and heterogeneity, and we also used PPO as an example to explore how to enhance the ability of a RL approach under entropy and heterogeneity. There exist some other RL algorithms, such as meta-reinforcement learning, as well as attention-based and model-based MASAC or LLM. One of our future research directions is to extend the proposed PPO-based approach to these RL algorithms. In addition, this paper focuses on memorization-based APT attacks. In our future work, we will investigate how to extend the work to other APT attacks. Moreover, we only carry out experiments in a simulation platform. In the future, we will consider how to investigate the capability of the proposed approach in realistic environments.

## Figures and Tables

**Figure 1 entropy-27-01238-f001:**
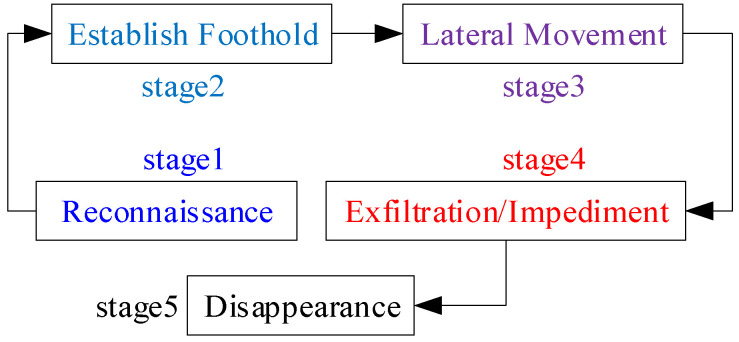
Schematic diagram of APT attack stages.

**Figure 2 entropy-27-01238-f002:**
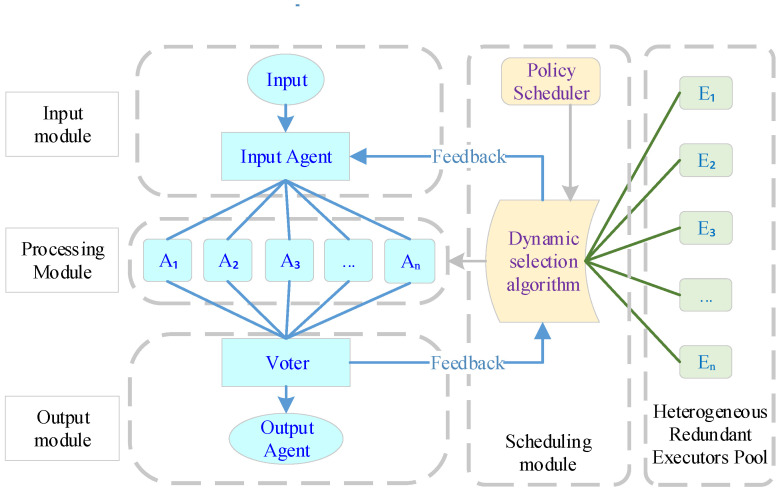
Dynamic Heterogeneous Redundancy model structures.

**Figure 3 entropy-27-01238-f003:**
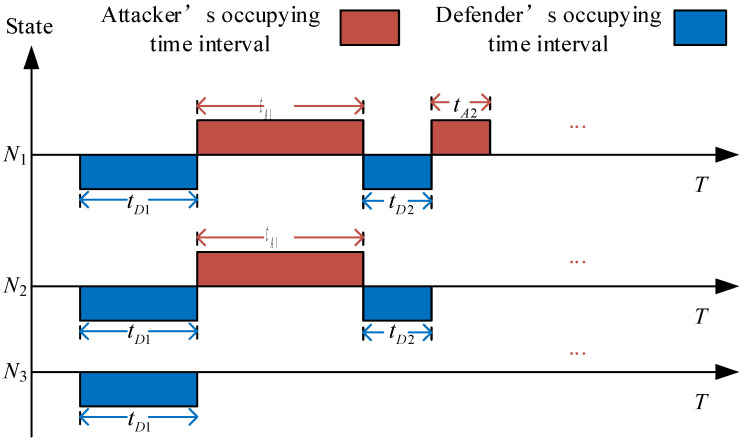
FlipIt-game attack–defense payoff diagram.

**Figure 4 entropy-27-01238-f004:**
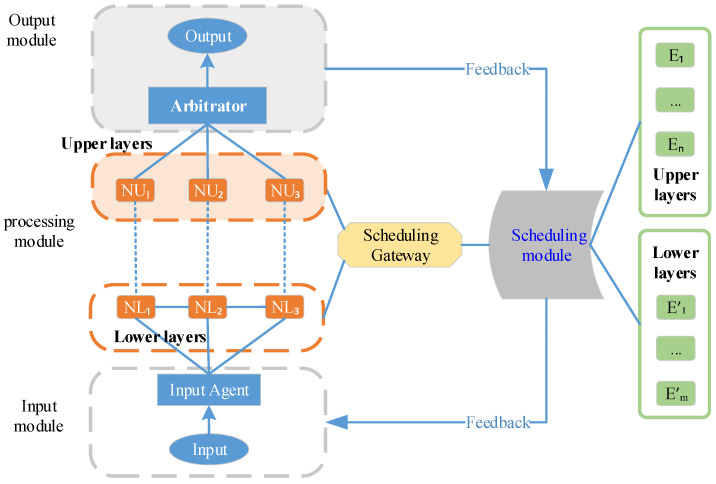
Schematic diagram of DDHR architecture with redundancy of 6.

**Figure 5 entropy-27-01238-f005:**
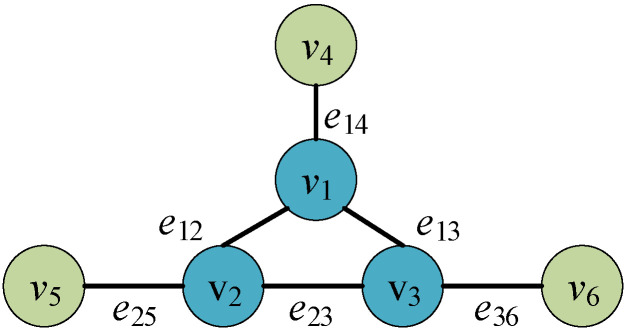
DDHR architecture attack graph network model with redundancy of 6.

**Figure 6 entropy-27-01238-f006:**
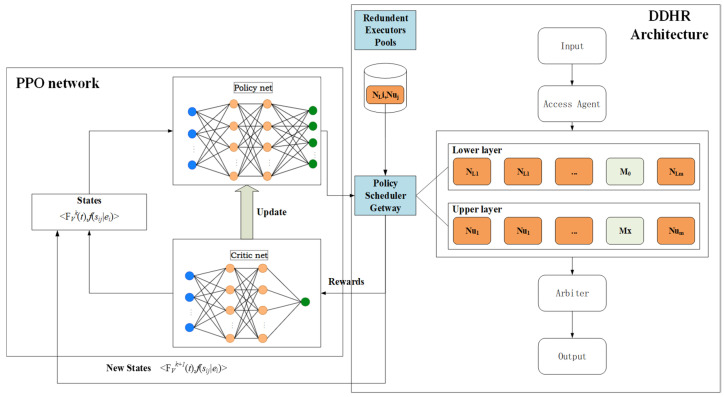
FlipIt-DDHR scheduling framework based on PPO-HE approach.

**Figure 7 entropy-27-01238-f007:**
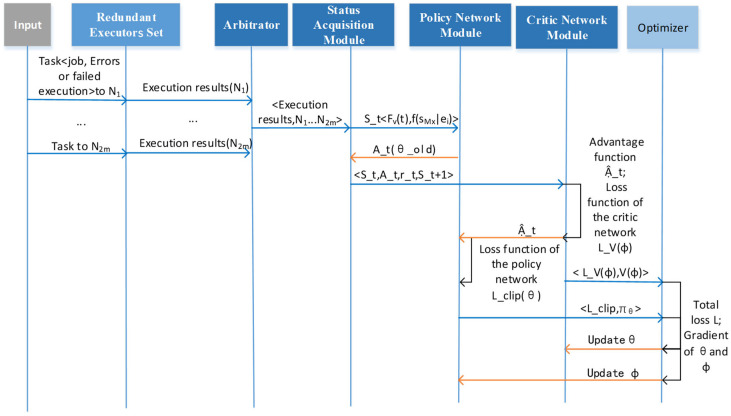
The approach workflow.

**Figure 8 entropy-27-01238-f008:**
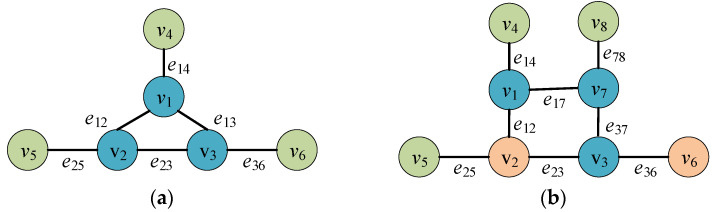
(**a**) Network connectivity and heterogeneous redundancy anchor point configuration diagram for DDHR architecture with redundancy 2*m* = 6, (**b**) with redundancy 2*m* = 8.

**Figure 9 entropy-27-01238-f009:**
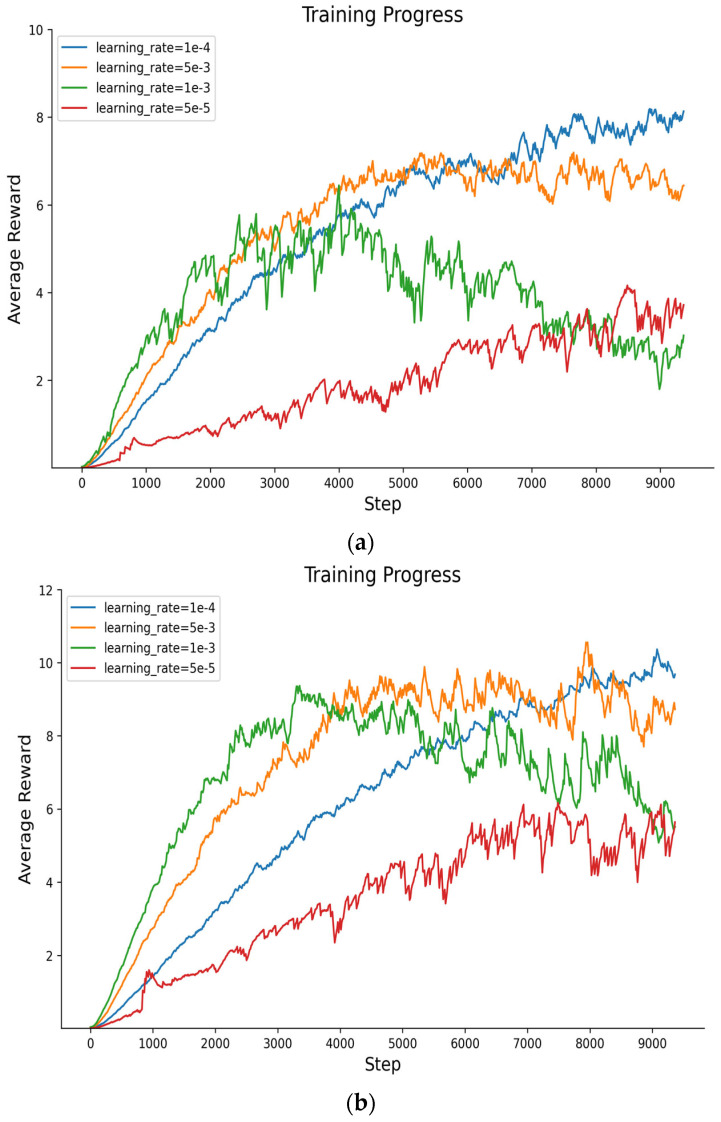
(**a**) The average reward for redundancy of 6 when learning rates are 1 × 10^−3^, 5 × 10^−3^, 1 × 10^−4^, and 5 × 10^−5^. (**b**) The average reward for redundancy of 8 when learning rates are 1 × 10^−3^, 5 × 10^−3^, 1 × 10^−4^, and 5 × 10^−5^.

**Figure 10 entropy-27-01238-f010:**
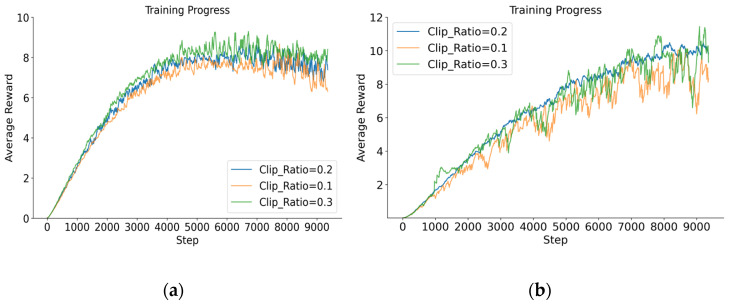
(**a**) The average reward for redundancy of 6 when clip ratio is set to 0.1, 0.2, and 0.3. (**b**) The average reward for redundancy of 8 when clip ratio is set to 0.1, 0.2, and 0.3.

**Figure 11 entropy-27-01238-f011:**
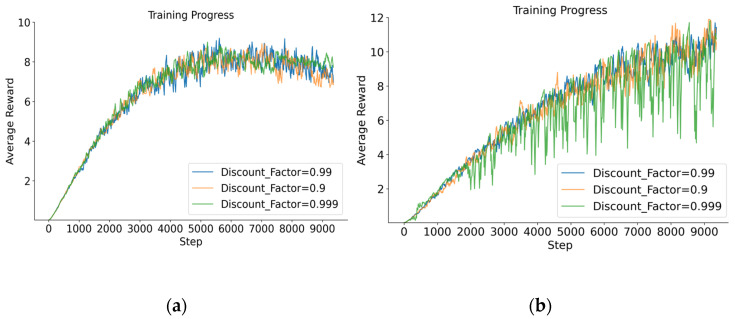
(**a**) The average reward for redundancy of 6 when discount factors are 0.9, 0.99, and 0.999. (**b**) The average reward for redundancy of 8 when discount factors are 0.9, 0.99, and 0.999.

**Figure 12 entropy-27-01238-f012:**
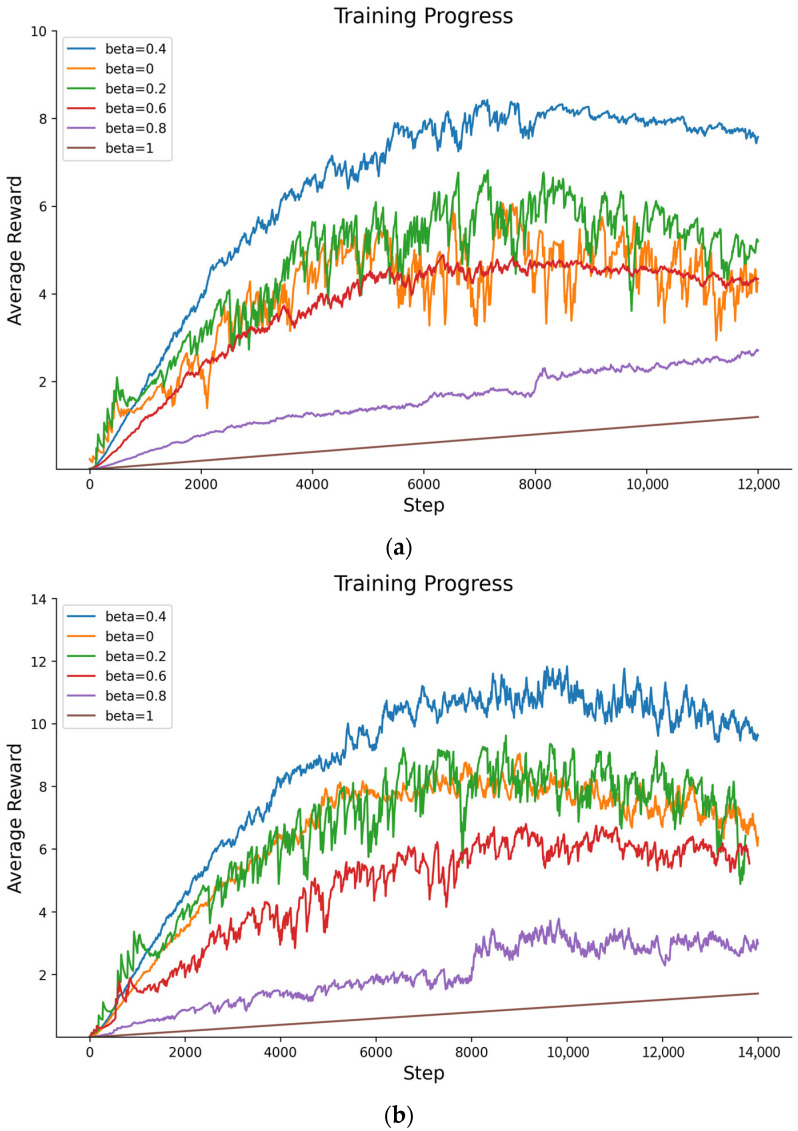
(**a**) The average reward for redundancy of 6 when β is 0, 0.2, 0.4, 0.6, 0.8, and 1. (**b**) The average reward for redundancy of 8 when β is 0, 0.2, 0.4, 0.6, 0.8, and 1.

**Figure 13 entropy-27-01238-f013:**
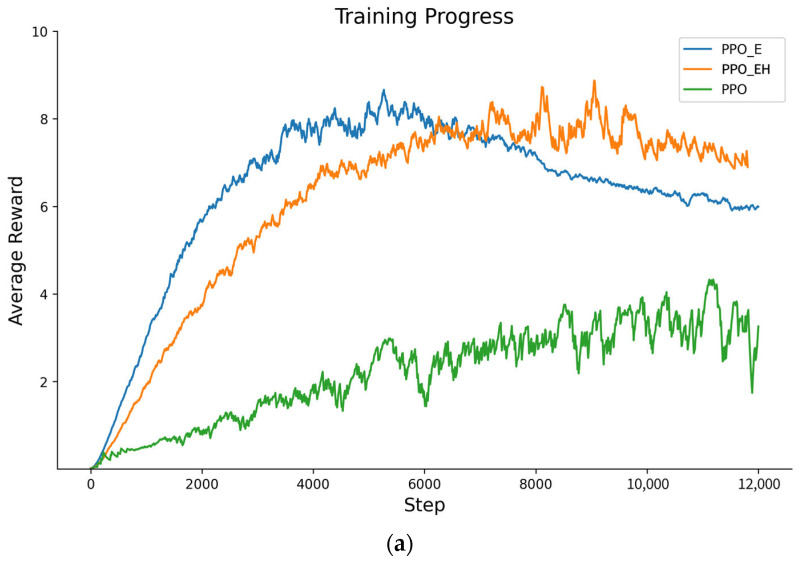
(**a**) The average reward for approaches PPO_EH, PPO_E, and PPO at redundancy of 6. (**b**) The average reward for approaches PPO_EH, PPO_E, and PPO at redundancy of 8.

**Figure 14 entropy-27-01238-f014:**
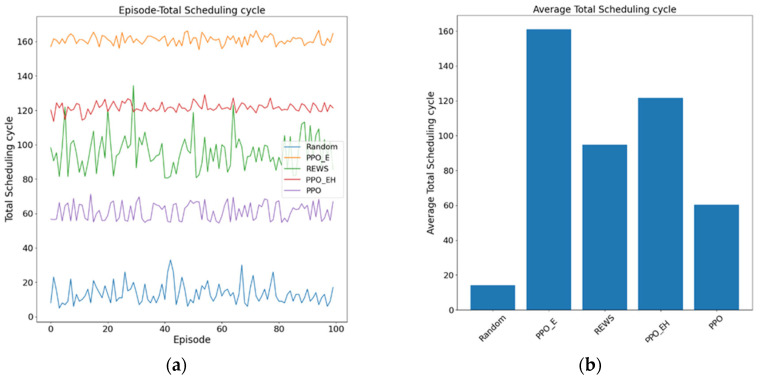
(**a**,**c**) Total scheduling cycle for the five approaches, PPO, PPO_E, PPO_EH, REWS, and Random at redundancy of 6 and 8. (**b**,**d**) The average scheduling cycle for the five approaches, PPO, PPO_E, PPO_EH, REWS, and Random at redundancy of 6 and 8.

**Figure 15 entropy-27-01238-f015:**
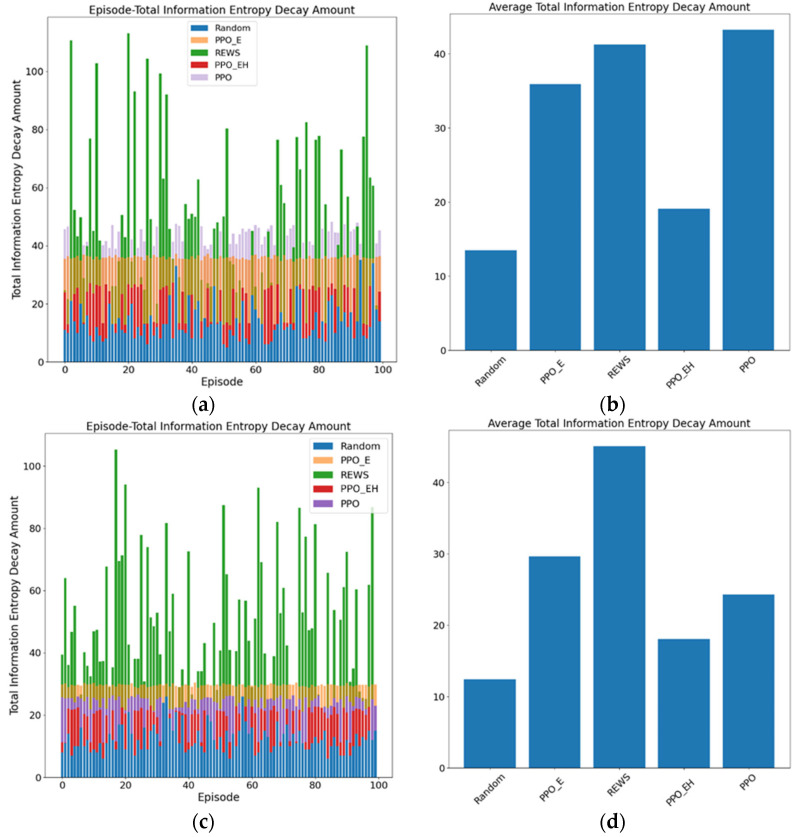
(**a**,**c**) Total information entropy decay amount for the five approaches: PPO, PPO_E, PPO_EH, REWS, and Random at redundancy of 6 and 8. (**b**,**d**) The average total information entropy decay amount for the five approaches, PPO, PPO_E, PPO_EH, REWS, and Random at redundancy of 6 and 8.

**Table 1 entropy-27-01238-t001:** Characteristics of different technical architectures deployed in centralized cloud environments.

Architecture Name	Redundancy Capacity	Fault Tolerance	Dynamic	Applicable Environment
DHR	High	High	√	Centralized
BFT	Mid	High	√	Distributed
FTR	Mid	Low	√	Centralized or Distributed

**Table 2 entropy-27-01238-t002:** Symbol definition.

Symbol	Definition
A	The game strategy
${\hat{A}}_{k}$	Advantage function
B	Total revenue from the occupation of public resources
b	Attack cost b ≥ 0
$C,c_{i}$	Set of node attack states, the i-th node in attack state
d	Defend cost d ≥ 0
$E,e_{ij}$	Weight set of connecting edges between redundant executors, the i-th and j-th redundant executor connectivity weights
e	Attacker’s base gain
$F_{i}(t)$	The i-th information entropy metrics of public resources in time period t
$F_{V}(t_{k})$	Total information entropy metrics of public resources in the k-th stage of time period t
$F_{V}(\Delta t_{k})$	Decay of information entropy metrics of public resources in the k-th stage of time period t
$F_{V}{\left(\Delta t_{k}\right)}_{ratio}$	Decay rate of information entropy metrics of public resources in the k-th stage of time period t
$f(s_{ij})$	Function of redundant executor average similarity affecting attack success probability in DHR Architecture
$G_{DDHR}$	DDHR architecture attack graph network model
$L^{clip}(\theta )$	When using the clip method, the loss function of the policy network
L(ϕ)	The loss function of the critic network
$M_{i}$	The i-th anchor redundant executor
m	Total number of upper/lower layer redundant executors
N, $N_{A}$, $N_{D}$, $N_{o}$	Denote all participants in the game, the attackers, the defenders, and neutral resources, respectively
n	Total number of redundant executors in the resource pool
$p_{i}(t)$	Probability of a successful attack on redundant executor i during time period t
q()	State transition probability function for transitioning to the next stage
$R_{D}$	Optimal payoff function for the defending side
S	State of equilibrium
$T,T^{\prime }$	Denote the executor scheduling time and attack time, respectively
Ts	Scheduling Cycle for Redundant executors
t	The time period required to occupy resources, t∈T
tr	Trajectory of executing scheduling strategy
$U_{A}$,$U_{D}$	Represent the attacker’s and defender’s payoff, respectively
$V,v_{i}$	Redundant executor node set, the i-th redundant executor node
*Z*	FlipIt game model
λ(t)	The coefficient of the proportion of total revenue occupied by the attacker during time period t
φ()	Correlation function between revenue and information entropy metrics
ε	Correlation coefficient between revenue and information entropy indicators
τ	Number of samples for discretization of information entropy metrics
ρ(t)	Attack distribution function on time interval t
γ	Discount factor for status transfer
$\omega (c_{i})$	Weight adjustment function for redundant executor adjudication
$\sigma (c_{i})$	Redundant executor scheduling function
$\kappa (s_{M_{X}})$	Setting action function for heterogeneous redundant executor anchor similarity
θ,ϕ	Parameters of policy network and critic network
$\pi_{\theta }$	Scheduling policy

**Table 3 entropy-27-01238-t003:** Experimental environment.

CPU	Intel Core i7 Ultra, Intel Corporation, Penang, Malaysia
Memory	64GB DDR5, frequency 4800 MHz and timing CL40, SK Hynix, Wuxi, China
Motherboard	Alienware Model 0P4K4P (Chipset: Intel HM770), Foxconn, Shenzhen, China
BIOS version	1.8.0
Storage	2 TB Samsung PM9A1 NVMe M.2 SSD (Interface: PCIe 4.0 x4), Samsung, Hwaseong, South Korea
Operating system	Microsoft Windows 11 Pro, Version 23H2 (OS Build 22631.3447)
Software environment	Python 3.13 + PyTorch 2.0.1

**Table 4 entropy-27-01238-t004:** Experimental parameter settings.

Parameters	Value
Total number of episodes	1.2 × 104, 1.4 × 10^4^
Number of slots in each episodes	1000
Actor and critic learning rates	1 × 10^−3^, 5 × 10^−3^, 1 × 10^−4^, 5 × 10^−5^
Actor network	12 × 64 × 64 × 2
Critic network	12 × 64 × 64 × 1
Discount factor γ	0.9, 0.99, 0.999
Clip ratio	0.1, 0.2, 0.3
PPO epoch	10
Learning optimizer	Adam
Probability of failure of redundant executors	0.1
Information entropy decay rate during training phase	0.01
Information entropy decay rate in the application phase of algorithm simulation	0.05
Attacker’s base reward eε	−0.5

**Table 5 entropy-27-01238-t005:** The average reward value (AR) for redundancy of 6 and 8 when learning rates are 1 × 10^−3^, 5 × 10^−3^, 1 × 10^−4^, and 5 × 10^−5^.

Learning Rate	1 × 10^−3^	5 × 10^−3^	1 × 10^−4^	5 × 10^−5^
2*m* = 6(AR)	3.340	5.338	6.242	3.107
2*m* = 8(AR)	6.668	7.169	7.875	5.791

**Table 6 entropy-27-01238-t006:** The average reward value (AR) for redundancy of 6 and 8 when clip ratio is set to 0.1, 0.2, and 0.3.

Clip Ratio	0.1	0.2	0.3
2*m* = 6(AR)	6.389	6.401	6.477
2*m* = 8(AR)	6.092	6.603	6.607

**Table 7 entropy-27-01238-t007:** The average reward value (AR) for redundancy of 6 and 8 when discount factor are 0.9, 0.99, and 0.999.

Discount Factor	0.9	0.99	0.999
2*m* = 6(AR)	6.362	6.502	6.383
2*m* = 8(AR)	6.347	6.633	6.210

**Table 8 entropy-27-01238-t008:** The average reward value (AR) Table for redundancy of 6 and 8 when β is 0, 0.2, 0.4, 0.6, 0.8, and 1.

*β*	0	0.2	0.4	0.6	0.8	1
2*m* = 6(AR)	4.415	4.882	6.056	3.487	1.490	0.601
2*m* = 8(AR)	6.657	7.103	8.324	4.551	2.209	0.655

**Table 9 entropy-27-01238-t009:** The average reward for approaches PPO_EH, PPO_E, and PPO at redundancy of 6 and 8.

	PPO_EH	PPO_E	PPO
2*m* = 6(AR)	6.245	6.530	2.872
2*m* = 8(AR)	8.261	8.655	4.002

**Table 10 entropy-27-01238-t010:** The average scheduling cycle and experimental variance for different approaches at redundancy levels of 6 and 8.

	Redundancy 2*m* = 6		Redundancy 2*m* = 8	
	Average Scheduling Cycle	Experimental Variance Scheduling	Average Scheduling Cycle	Experimental Variance Scheduling
Random	13.43	32.85	12.52	19.71
REWS	95.84	106.87	86.73	52.29
PPO	61.21	25.24	30.72	14.05
PPO_E	161.07	6.46	134.69	2.98
PPO_EH	121.65	7.29	103.73	8.85

**Table 11 entropy-27-01238-t011:** The information entropy decay amount and decay rate for different approaches under redundancy of 6 and 8.

	Redundancy 2*m* = 6		Redundancy 2*m* = 8	
	Information Entropy Decay Amount	Information Entropy Decay Rate	Information Entropy Decay Amount	Information Entropy Decay Rate
Random	12.50	0.93	12.10	0.97
REWS	41.22	0.43	45.06	0.52
PPO	43.21	0.71	24.27	0.79
PPO_E	35.91	0.22	29.64	0.22
PPO_EH	19.08	0.16	18.04	0.17

## Data Availability

No new data were created or analyzed in this study. Data sharing is not applicable to this article.

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
