# Peer review of "Resisting Memorization-Based APT Attacks Under Incomplete Information in DDHR Architecture: An Entropy-Heterogeneity-Aware RL-Based Scheduling Approach"

_entropy, 2025, doi:10.3390/e27121238_

Round 1
Reviewer 1 Report
Comments and Suggestions for Authors
The reviewed article addresses the issue of countering memorization-based APT attacks in train passenger service systems. The presented method focuses on solving the problem of the inability to accurately estimate the planning time and the lack of an intelligent defensive planning method capable of counteracting memorization-based attacks. The research includes, among others, a formal model of the defender's payoff optimization problem under incomplete information, an attack time estimation method, and a proximal policy optimization algorithm. The following are comments that may help improve the quality of the final article.
Comment 1: Tables 2 and 3 (pages 4-5) could be moved to the section where the defined symbols are used. Currently, the tables are in the "Related Works" section, where the results of other available studies are analyzed. A justification for separating the symbol definitions into two different tables, Tables 2 and 3, which share the same title, should also be provided.
Comment 2. Research Methodology: Section 3 contains the formulation of the payoff optimization problem under conditions of incomplete information. This section covers the formal formulation of the mathematical model and falls within the scope of theoretical and model-based research. After the Related Works section, it is worth considering adding a new Research Methodology section. In this section, the scientific apparatus should be defined, including: the main research problem, specific problems, as well as research methods, techniques, and tools. The adopted research methods, techniques, and tools should be related to the scientific framework appropriate to the scientific discipline in which the research is conducted.
Comment 3. The description of the experimental research environment (lines 641-645) should be expanded to include additional information: RAM (standard, frequency, latency), motherboard (mode, BIOS version), hard drive (type, interface), operating system (compilation number). It is worth presenting the full technology stack of the experimental research environment in notation consistent with software engineering guidelines.
Comment 4. In section 5.1.2 it is worth justifying the selection of parameter values adopted in experimental studies (Table 4).
Comment 5. The article should clearly indicate the limitations of the presented method.
Comment 6. The Conclusions section should suggest directions for further research.
Author Response
Comment 1: Tables 2 and 3 (pages 4-5) could be moved to the section where the defined symbols are used. Currently, the tables are in the "Related Works" section, where the results of other available studies are analyzed. A justification for separating the symbol definitions into two different tables, Tables 2 and 3, which share the same title, should also be provided.
Author response: We really appreciate this valuable comment from the reviewer, which makes the manuscript much easier to understand. There are many symbols to define, which can not be in a page. Then, we use two tables. We apologized for the confuse caused. In the revised manuscript, we use a table for the definition of all symbols.
Comment 2. Research Methodology: Section 3 contains the formulation of the payoff optimization problem under conditions of incomplete information. This section covers the formal formulation of the mathematical model and falls within the scope of theoretical and model-based research. After the Related Works section, it is worth considering adding a new Research Methodology section. In this section, the scientific apparatus should be defined, including: the main research problem, specific problems, as well as research methods, techniques, and tools. The adopted research methods, techniques, and tools should be related to the scientific framework appropriate to the scientific discipline in which the research is conducted.
Author response: We really appreciate this valuable comment from the reviewer, which makes the manuscript much easier to understand. We definitely agree with the reviewer about adding research methodology. We revised the last paragraph by adding this methodology.
Comment 3. The description of the experimental research environment (lines 641- 645) should be expanded to include additional information: RAM (standard, frequency, latency), motherboard (mode, BIOS version), hard drive (type, interface), operating system (compilation number). It is worth presenting the full technology stack of the experimental research environment in notation consistent with software engineering guidelines.
Author response: We really appreciate this valuable comment from the reviewer, which makes the manuscript more comprehensive. We addressed this comment by adding a detailed description of the experimental environment at the beginning of Section 5.
Comment 4. In section 5.1.2 it is worth justifying the selection of parameter values adopted in experimental studies (Table 4)
Author response: We really appreciate this valuable comment from the reviewer, which makes the manuscript more comprehensive. In Section 5.2, we added three additional sets of experiments comparing parameters related to clip ratio, discount factor, and reward function weighting, in addition to the learning rate parameter comparison experiments. The values of the first parameters are set according to literatures related to PPO algorithm. The values of the last four parameters are set based on our experiments. We justified the selection of parameter values at the end of the first paragraph of Section 5.1.2. We also added experiment results in Section 5.2.2-5.2.4 to show impact of paraments values on metrics in the revised manuscript.
Comment 5. The article should clearly indicate the limitations of the presented method.
Author response: We really appreciate this valuable comment from the reviewer, which makes the manuscript more comprehensive. We identified three limitations with the paper work the last paragraph of the paper. The first is the lack of comparing with the other RL algorithms. The second is that we only consider memorization-based APT attack. The last one is the evaluation environment.
Comment 6. The Conclusions section should suggest directions for further research.
Author response: We really appreciate this valuable comment from the reviewer, which makes the manuscript more comprehensive. We added future researches in the last paragraph of the paper.
Reviewer 2 Report
Comments and Suggestions for Authors
The work is very interesting and important. Some suggestions can be considereded:
1. In the PPO_EH approach, the paper mentions integrating quantifiable information entropy and heterogeneity of DDHR redundant executors into the PPO algorithm. However, the specific mechanism of how these two factors are embedded into the PPO’s update process (e.g., whether they are added to the state input, policy loss function, or advantage estimation) is not clearly elaborated. Could the authors provide more technical details on how entropy and heterogeneity metrics interact with PPO’s core components (e.g., clip loss, GAE-based advantage calculation) to optimize scheduling strategies?
2. The reward function of PPO_EH combines the defender’s payoff and the effective reward of the DDHR architecture, with α and β as normalization parameters. However, the paper does not explain how α and β are determined, nor does it verify the impact of different weight allocations between entropy-related terms and defense payoff terms on PPO’s convergence and final defense effectiveness. Could the authors provide ablation experiments or parameter tuning results to illustrate the contribution of each component in the reward function to PPO_EH’s performance?
3. The experimental results show that a learning rate of 1×10⁻⁴ yields the best performance for PPO_EH. However, the paper only compares four learning rate values (1×10⁻³, 5×10⁻³, 1×10⁻⁴, 5×10⁻⁵) without providing a comprehensive hyperparameter sensitivity analysis. For critical PPO hyperparameters such as the clip ratio (set to 0.2) and discount factor (set to 0.99), are there any experiments to verify why these specific values are chosen? Would adjusting these hyperparameters affect the stability and convergence speed of PPO_EH when countering memorization-based attacks?
4. The reward function of PPO_EH combines the defender’s payoff and the effective reward of the DDHR architecture, with α and β as normalization parameters. However, the paper does not explain how α and β are determined, nor does it verify the impact of different weight allocations between entropy-related terms and defense payoff terms on PPO’s convergence and final defense effectiveness. Could the authors provide ablation experiments or parameter tuning results to illustrate the contribution of each component in the reward function to PPO_EH’s performance?
5. Some other RL algorithms can be investigated, such as Meta-reinforcement learning, attention-based , model-based MASAC or LLM for Malware Propagation or similar attacks. These new methods can be compared simply.
Author Response
Comment 1: In the PPO_EH approach, the paper mentions integrating quantifiable information entropy and heterogeneity of DDHR redundant executors into the PPO algorithm. However, the specific mechanism of how these two factors are embedded into the PPO’s update process (e.g., whether they are added to the state input, policy loss function, or advantage estimation) is not clearly elaborated. Could the authors provide more technical details on how entropy and heterogeneity metrics interact with PPO’s core components (e.g., clip loss, GAE-based advantage calculation) to optimize scheduling strategies?
Author response: We really appreciate this valuable comment from the reviewer and we provided a detailed description in Section 4.3 regarding the initialization of information entropy, anchor position redundancy similarity weights to the state space. Additionally, we elaborated on how the model state is transformed into the state value function through the network during the calculation of the advantage function.
Comment 2: The reward function of PPO_EH combines the defender’s payoff and the effective reward of the DDHR architecture, with α and β as normalization parameters. However, the paper does not explain how α and β are determined, nor does it verify the impact of different weight allocations between entropy-related terms and defense payoff terms on PPO’s convergence and final defense effectiveness. Could the authors provide ablation experiments or parameter tuning results to illustrate the contribution of each component in the reward function to PPO_EH’s performance?
Author response: We really appreciate this valuable comment from the reviewer, which makes the revised manuscript more comprehensive. In Section 5.2.4, we added additional experiment comparing parameters related to reward function weighting, in addition to the learning rate parameter comparison experiments.
Comment 3: The experimental results show that a learning rate of 1×10⁻⁴ yields the best performance for PPO_EH. However, the paper only compares four learning rate values (1×10⁻³, 5×10⁻³, 1×10⁻⁴, 5×10⁻⁵) without providing a comprehensive hyperparameter sensitivity analysis. For critical PPO hyperparameters such as the clip ratio (set to 0.2) and discount factor (set to 0.99), are there any experiments to verify why these specific values are chosen? Would adjusting these hyperparameters affect the stability and convergence speed of PPO_EH when countering memorization-based attacks?
Author response: We really appreciate this valuable comment from the reviewer, which makes the revised manuscript more comprehensive. In Section 5.2.2 and 5.2.3, we added additional experiments for investigating the impact of parameters of clip ratio and discount factor, in addition to the parameter of learning rate.
Comment 4: The reward function of PPO_EH combines the defender’s payoff and the effective reward of the DDHR architecture, with α and β as normalization parameters. However, the paper does not explain how α and β are determined, nor does it verify the impact of different weight allocations between entropy-related terms and defense payoff terms on PPO’s convergence and final defense effectiveness. Could the authors provide ablation experiments or parameter tuning results to illustrate the contribution of each component in the reward function to PPO_EH’s performance?
Author response: We really appreciate this valuable comment from the reviewer, which makes the revised manuscript more comprehensive. In Section 5.2.4, we added additional experiment comparing parameters related to reward function weighting, in addition to the learning rate parameter comparison experiments.
Comment 5: Some other RL algorithms can be investigated, such as Meta-reinforcement learning, attention-based, model-based MASAC or LLM for Malware Propagation or similar attacks. These new methods can be compared simply.
Author response: We really appreciate this valuable comment from the reviewer, which makes the revised manuscript more comprehensive. We definitely agreed with the reviewer about this comparison. This paper aims to explore the importance of entropy and heterogeneity, and we also used PPO as an example to explore how to enhance the ability of a RL approach under entropy and heterogeneity. We leave the experiment comparison of PPO with other RL algorithms for future works. We added future researches in the last paragraph of the paper.
Round 2
Reviewer 2 Report
Comments and Suggestions for Authors
No further problems.